# OML: A Primitive for Reconciling Open Access with Owner Control in AI Model Distribution

## Abstract

The current paradigm of AI model distribution presents a fundamental dichotomy: models are either closed and API-gated, sacrificing transparency and local execution, or openly distributed, sacrificing monetization and control. We introduce **OML** (Open-access, Monetizable, and Loyal AI Model Serving), a primitive that enables a new distribution paradigm where models can be freely distributed for local execution while maintaining cryptographically enforced usage authorization. We are the first to introduce and formalize this problem, introducing rigorous security definitions tailored to the unique challenge of white-box model protection: *model extraction resistance* and *permission forgery resistance*. We prove fundamental bounds on the achievability of OML properties and characterize the complete design space of potential constructions, from obfuscation-based approaches to cryptographic solutions. To demonstrate practical feasibility, we present OML 1.0, a novel OML construction leveraging AI-native model fingerprinting coupled with crypto-economic enforcement mechanisms. Through extensive theoretical analysis and empirical evaluation, we establish OML as a foundational primitive necessary for sustainable AI ecosystems. This work opens a new research direction at the intersection of cryptography, machine learning, and mechanism design, with critical implications for the future of AI distribution and governance.

## 1 Introduction

Artificial Intelligence (AI) is advancing at an incredible pace, reshaping diverse fields from household robotics iRobot (2023); Dynamics (2023) and superhuman game-playing Silver et al. (2017a;b; 2018) to intricate formal mathematical reasoning DeepMind (2024), protein structure elucidation Jumper et al. (2021); Evans et al. (2021), accelerated drug discovery Boström et al. (2018); Strokach et al. (2020); Schneider et al. (2020), and novel mathematical exploration Romera-Paredes et al. (2024). The emergence of powerful generative models like GPT series OpenAI (2023b); Bubeck et al. (2023), OpenAI `o3` OpenAI (2024); Jaech et al. (2024), and DeepSeek `R1` Guo et al. (2025) marks a watershed moment, heralding their potential to fundamentally transform human endeavors.

Yet, the rapid advancement of artificial intelligence has created an unprecedented challenge in model distribution. Current approaches force an unnecessary tradeoff between accessibility and control, limiting both innovation and sustainable development of AI systems. This paper introduces and formalizes OML, a primitive that reconciles these conflicting requirements.

### 1.1 The Fundamental Distribution Problem

Modern AI development, responsible for the ubiquitous systems we see today, has been significantly shaped by open-source contributions. Until recently, core libraries and powerful models such as BERT Devlin et al. (2018) and early iterations of GPT OpenAI (2023b); Bubeck et al. (2023) were openly available. However, as AI matured and its profound economic potential became evident, many large companies that initially embraced open development have transitioned to closed strategies. These entities have often geared their efforts towards establishing significant positions in the AI economy, with a model where others act as high-level users of the AI they build OpenAI (2023a); Forefront (2023); Labs (2023). Consequently, AI is currently delivered to users predominantly via the following two dominant paradigms, each with critical limitations.

**Closed API Services**: In this paradigm, AI models are primarily accessed through public APIs OpenAI (2023a); Forefront (2023); Labs (2023). Platforms like OpenAI's GPT and Anthropic's Claude maintain complete control over model execution, enabling monetization and usage governance. Such centralized, closed services offer benefits like scalability and the implementation of certain safety measures, including content moderation and misuse prevention. Conversely, this approach can lead to monopolization, rent-seeking behaviors, and significant privacy concerns. Users also lack ultimate control over the paid service, as model owners can arbitrarily filter inputs, modify outputs, or change the underlying model without direct user consent. While options for fine-tuning closed models may exist, such customization is typically constrained by the limitations of the provided API. Users cannot verify model behavior, ensure data privacy, or maintain operational independence.

**Open-weight distribution**: In this paradigm, creators release model weights, often with their architectures, allowing users to download and run inference locally. Platforms like Hugging Face enable unrestricted model distribution, providing transparency, local execution, and modification capabilities. This grants users full control over model selection, inference processes, and the ability to build upon these models (e.g., through fine-tuning) and compose them with other AI systems. Meta's Llama model series Touvron et al. (2023a), DeepSeek Guo et al. (2025), and the wide array of models available on platforms like Hugging Face exemplify this approach. However, once models are released in this manner, creators lose direct control over their subsequent use, cannot enforce safety constraints and prevent unethical applications, and lack mechanisms for sustainable monetization. This creates a tragedy of the commons where innovation is undersupported.

This dichotomy is not merely a business model choice but represents a **fundamental technical limitation** in our current infrastructure. We lack the primitives necessary to enable models that are simultaneously open-access for local execution and controlled for authorized and ethical usage.

## 1.2 OML: Technical Requirements for Ideal Resolution

While both closed and open models offer distinct advantages, we pursue maximal openness comparable to current open-weight distributions, augmented with mechanisms for owner control. An ideal solution must therefore reconcile three seemingly contradictory properties, which we term "OML":

- **[O] Open-access**. Models must be freely distributable for local execution, analogous to compiled binaries where functionality is accessible while implementation details remain protected. Once distributed, models become immutable artifacts independent of their creators, enabling user data privacy, consistent service quality, and on-premise deployment.

- **[M] Monetizable**. The framework enables model owners to capture economic value through granular, per-inference authorization mechanisms. Each model invocation requires individual permission from model owners, and an economic Nash Equilibrium, where rational users obtain proper authorization (such as purchasing access tokens) rather than attempting circumvention, guarantees that the all agreements and licenses are enforced.

- **[L] Loyal**. Models must technically enforce owner-defined policies through pre-hoc authorization verification. The model produces high-utility outputs exclusively when presented with valid, cryptographically-bound permissions, ensuring compliance with safety and ethical constraints before computation rather than through legal remedies after violation.

Note that, while monetizability and loyalty both enable governance, they address fundamentally different requirements. Monetizability concerns economic sustainability and can leverage post-hoc mechanisms such as collateral-backed compliance. Loyalty demands pre-hoc technical enforcement to prevent generation of harmful or policy-violating outputs before they occur.

This challenge becomes particularly acute under white-box access conditions where users possess complete visibility into model weights, architecture, and computation flow. Unlike traditional software where obfuscation can leverage discrete execution paths and control flow complexity, neural networks present continuous, differentiable computations that resist conventional protection mechanisms. Any solution must therefore leverage properties unique to machine learning systems rather than adapting existing software protection paradigms.

## 1.3 OUR CONTRIBUTIONS

**Core contribution.** We are the first to identify and formalize the OML challenge, a primitive that enables AI models to be distributed openly while maintaining cryptographically enforced usage control under white-box access. This paper establishes the foundations for OML through:

1. **Problem Formalization**: We are the first to provide rigorous definition of requirements for reconciling open access with owner control, with three properties (Open-access, Monetizable, Loyal) and quantifiable metrics($\epsilon_{utility}, \epsilon_{robust}, \epsilon_{overhead}$) for evaluating solutions.

2. **Security Framework**: We establish novel security games for white-box model protection (model extraction resistance, permission forgery resistance) where adversaries have complete visibility into weights and computation, establishing security standards for OML realization.

3. **Design Space Characterization**: We analyze a wide array of potential approaches (obfuscation, TEEs, cryptography) with their fundamental tradeoffs and theoretical bounds.

4. **Feasibility Demonstration**: We introduce and instantiate OML 1.0, a practical construction using AI-native fingerprinting that achieves "next-day security" with negligible $\epsilon_{overhead}$. Experiments and empirical validation further demonstrates feasibility of our approach.

**Our primary contribution is establishing OML as a well-defined primitive and demonstrating its feasibility, thereby opening a new research direction at the intersection of cryptography, machine learning, and mechanism design.** The challenge of protecting neural networks under white-box access while preserving utility represents one of the most difficult problems in cryptography and AI, requiring fundamentally new techniques beyond classical approaches. We identify critical open problems spanning theoretical questions (tight complexity bounds, connections to program obfuscation, optimal constructions) and practical challenges (robustness under fine-tuning, compositional security, efficient real-time enforcement).

The full realization of OML will likely require years of sustained research effort from the community, comparable to the decades-long development of practical homomorphic encryption or secure multi-party computation. By providing the first formal framework and proving initial feasibility through OML 1.0, we aim to catalyze the long-term research program necessary to advance OML from primitive concept to robust, deployable infrastructure for AI distribution.

## 1.4 SIGNIFICANCE AND IMPACT

Through the OML primitive, we can address critical safety and privacy challenges faced in AI development today, while enabling a more collaborative ecosystem for AI development tomorrow:

**Privacy-Preserving Local Execution**: OML enables organizations handling sensitive data, e.g. healthcare providers, financial institutions, government agencies, to deploy state-of-the-art models without exposing confidential information to external APIs. Medical institutions can run diagnostic models on patient data locally while still compensating model creators. Financial firms can apply fraud detection models without sharing transaction patterns. This solves the current impossibility where organizations must choose between using inferior models or violating privacy requirements.

**Technical Enforcement of Ethical and Legal Usage**: By requiring pre-hoc cryptographic authorization for each inference, OML provides robust mechanisms for enforcing safety policies and any signed user agreements on open-weight models that currently rely solely on terms of service today.

**Enabling a More Colloborative Ecosystem for AI**: Most fundamentally, OML creates the technical infrastructure for distributed AI development where every contribution, from initial training to incremental improvements, can be tracked and rewarded. Figure 1 illustrates how OML transforms the current one-way distribution into a sustainable ecosystem where usage generates returns for all contributors, creating economic incentives for continuous improvement rather than one-time model releases. Collaboration with positive feedback loop and proper incentives, rather than monopolization, guarantees that AI development is on the right track that benefits the humanity.

These three impacts are mutually reinforcing: privacy-preserving local execution expands the market for AI models; license enforcement makes broader deployment responsible; and collaborative development efforts ensure sustainable innovation. Together, they address the fundamental limitations preventing AI from achieving its potential as a broadly beneficial technology.

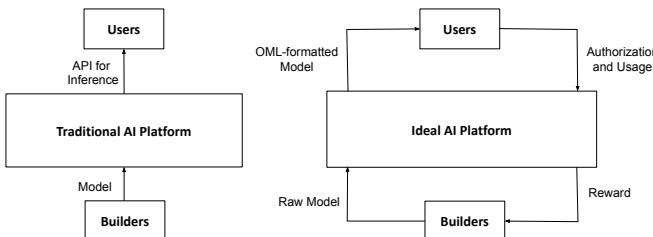

Figure 1: OML enables transition from one-way model distribution to bidirectional value flow. Left: Current paradigm where models are distributed without feedback or compensation mechanisms. Right: OML-enabled ecosystem where usage generates returns for all contributors, incentivizing continuous collaborative improvement.

## 2 THE OML PRIMITIVE: FORMAL DEFINITION

The transformation of AI models from static artifacts to dynamic, controllable assets requires a fundamental reconceptualization of how we distribute and govern computational intelligence. In this section, we formalize the *Open-access, Monetizable, and Loyal (OML)* primitive, a framework that enables white-box model distribution while preserving ownership rights and enforcing usage policies.

### 2.1 PROPERTIES AND DESIGN SPACE

Consider an AI model $M$ as an economic asset: valuable, replicable, and vulnerable to unauthorized extraction once distributed. The OML primitive transforms this vulnerable asset into a controlled artifact that retains its utility while coupling high-quality outputs to owner authorization.

We begin by establishing our notation framework, which we will use throughout this section.

Table 1: Notation and Core Components of the OML Framework

| Symbol | Description |
|---|---|
| $M : \mathcal{X} \to \mathcal{Y}$ | Original model mapping inputs to outputs |
| $M_{\mathrm{oml}}$ | OML-formatted model with embedded authorization |
| $h : \mathcal{X} \to \mathcal{H}$ | Input-binding transform (e.g., cryptographic commitment) |
| $\sigma : \mathcal{H} \times \mathcal{K}_{\mathrm{own}} \to \mathcal{P}$ | Permission token generator |
| $k_{\mathrm{own}}$ | Owner's secret key; $vk_{\mathrm{own}}$ denotes optional public verifier |
| $p_x = \sigma(h(x), k_{\mathrm{own}})$ | Permission token cryptographically bound to input $x$ |
| $d(\cdot, \cdot)$ | Task-appropriate distance or divergence metric |
| $\epsilon_{\mathrm{utility}}$ | Maximum fidelity loss on authorized queries |
| $\epsilon_{\mathrm{robust}}$ | Minimum degradation on unauthorized queries |
| $\epsilon_{\mathrm{overhead}}$ | Relative computational overhead bound |

With this notation established, we can now formally define the OML transformation, a process that embeds authorization logic so deeply within the model's computational graph that removing it becomes computationally equivalent to retraining the model from scratch.

**Definition 2.1 (OMLized Model).** Given an original model $M : \mathcal{X} \to \mathcal{Y}$, an OMLization process

$$\mathrm{OMLize}(M; h, \sigma, \mathrm{params}) \longrightarrow M_{oml},$$

produces a locally executable artifact that operates on input-token pairs $(x, p)$. For each input $x \in \mathcal{X}$, authorization requires a valid token $p_x = \sigma(h(x), k_{own})$ computed with owner's secret key $k_{own} \in \mathcal{K}_{own}$. Informally, $M_{oml}$ behaves as $M$ on authorized inputs and degrades otherwise.

This definition captures the essence of controlled distribution: the model remains functionally accessible but computationally gated. To understand how this works in practice, we present the idealized OML workflow, which demonstrates how authorization, utility preservation, and security enforcement interact.

**Promises of an Ideal OML.** Let $d : \mathcal{Y} \times \mathcal{Y} \to \mathbb{R}_{\geq 0}$ denote a distance metric and $T(F, z)$ the computational cost of evaluating function $F$ at input $z$. A correct OMLization satisfies:

1. **Authorization:** Users submit $h(x)$ to owner $\Pi_{\mathcal{O}}$; if approved, they receive $p_x$ and query $(x, p_x)$.

2. **Fidelity:** $d(M_{oml}(x, p_x), M(x)) \leq \epsilon_{\text{utility}}$, ensuring preservance of the model's core capabilities.

3. **Protection:** For invalid $p$, $d(M_{oml}(x, p), M(x)) > \epsilon_{\text{robust}}$ with $\epsilon_{\text{robust}} > \epsilon_{\text{utility}}$.

4. **Overhead:** $T(M_{oml}, (x, p_x)) \leq (1 + \epsilon_{\text{overhead}}) T(M, x)$, preserving practical deployability.

These four promises collectively define what we term the *quality profile* $(\epsilon_{\text{utility}}, \epsilon_{\text{robust}}, \epsilon_{\text{overhead}})$, a quantitative characterization of an OMLization's effectiveness. And the instantiation requires careful design of three interconnected components that form the technical foundation of any OML construction listed below. We also depict a high-level OMLization process in Algorithm 1.

- an ownership key $k_{\text{own}}$ (either cryptographic or AI-native);
- a binding/permission token issuance mechanism $(h, \sigma)$, optionally exposing $vk_{\text{own}}$;
- *verifier entanglement*, which couples authorization to critical computations so that the high-utility pathway is reachable only under valid authorization.

---

**Algorithm 1:** OMLIZE: Transforming Models into Controlled Artifacts

1: **Input:** Original model $M$, binding function $h$, token scheme $\sigma$, public parameters
2: **Output:** Controlled artifact $M_{\text{oml}}$
3: **Step 1:** Embed verifier $\alpha : \mathcal{X} \times \mathcal{P} \to \{0, 1\}$ that validates tokens against input commitments
4: **Step 2:** Entangle $\alpha$ within $M$'s critical paths to construct $F$ such that:
    (i) Valid authorization: $\alpha(x, p_x) = 1 \Rightarrow F(x, p_x) \approx M(x)$
    (ii) Invalid tokens: $\alpha(x, p) \neq 1 \Rightarrow F(x, p)$ yields degraded/noisy output
5: **Step 3:** Optionally expose $vk_{\text{own}}$ for public verification capability
6: **return** $M_{\text{oml}}(x, p) = F(x, p)$

---

## 2.2 SECURITY GUARANTEES AND ADVERSARIAL MODEL

The security of OML must be analyzed under the assumption of white-box access where adversaries can inspect, modify, and experiment with $M_{oml}$ arbitrarily. This threat model reflects the reality where OMLized models may be controlled by potentially adversarial users.

**Adversary Model.** We model adversaries as probabilistic polynomial-time (PPT) algorithms $\mathcal{A}$ with

- Complete white-box access to $M_{oml}$, including all parameters and computation graphs
- Oracle access to an authorization service $\Pi_{\mathcal{O}}$ for up to $N$ queries
- The resulting knowledge base $\mathcal{D}_{known} = \{(x_i, p_{x_i}, y_i)\}_{i=1}^{N}$ where $y_i = M_{oml}(x_i, p_{x_i})$

**Security Goal.** Against such adversaries, two fundamental hardness properties should hold:

**Requirement 2.1** (**Model Extraction Resistance**)**.** In experiment $\text{Expt}_{\mathcal{A}}^{\text{ME}}$:

(1) $\mathcal{A}$ receives $M_{oml}$ and oracle access to $\mathcal{P}_{\mathcal{O}}$ for $N$ queries; (2) $\mathcal{A}$ outputs a stand-alone model $M'$;

(3) a fresh $x^* \sim \mathcal{D}_{\mathcal{X}}$ is drawn with $x^* \notin \{x_i\}$; (4) $\mathcal{A}$ *wins* if $d(M'(x^*), M(x^*)) \leq \epsilon_{utility}$.

The scheme is $(t, N, \epsilon_{ME})$-extraction-resistant if every PPT $\mathcal{A}$ running in time $t$ wins with probability at most $\epsilon_{ME}(t, N)$. Informally, any adversary cannot replicate a functionally equivalent model that bypasses authorization within reasonable cost.

**Requirement 2.2** (**Permission Forgery Resistance**)**.** In experiment $\text{Expt}_{\mathcal{A}}^{\text{PF}}$:

(1) $\mathcal{A}$ receives $M_{oml}$ and oracle access to $\mathcal{P}_{\mathcal{O}}$ for $N$ queries;

(2) a fresh $x^* \sim \mathcal{D}_{\mathcal{X}}$ is revealed with $x^* \notin \{x_i\}$;

(3) $\mathcal{A}$ outputs $p^*$; (4) $\mathcal{A}$ *wins* if $d(M_{oml}(x^*, p^*), M(x^*)) \leq \epsilon_{utility}$.

The scheme is $(t, N, \epsilon_{PF})$-forgery-resistant if every PPT $\mathcal{A}$ running in time $t$ wins with probability at most $\epsilon_{PF}(t, N)$. Informally, adversaries cannot generate valid tokens for unauthorized inputs.

These requirements must withstand a diverse threat landscape:

1. **Surgical Extraction:** Adversaries employ network surgery techniques Raiman et al. (2019); Guo et al. (2016) to identify and excise authorization logic while preserving model functionality.

2. **Runtime Manipulation:** Fault injection or state tampering forces the internal verifier to accept invalid tokens, bypassing authorization checks without modifying the model itself.

3. **Cryptanalytic Forgery:** Adversaries attempt to forge valid tokens through: (a) exploiting implementation vulnerabilities in $\sigma$, (b) recovering the secret key $k_{own}$ from side channels, or (c) training surrogate functions $\hat{\sigma} : x \mapsto p_x$ using $\mathcal{D}_{known}$.

**The Failure of Naive Approaches.** To illustrate why sophisticated entanglement is necessary, consider a naive wrapper design with a cryptographic digital signature scheme:

$$M_{oml}(x, p) := \begin{cases} M(x) & \text{if } \text{Verify}_{vk_{own}}(h(x), p) = \text{true} \\ \bot & \text{otherwise} \end{cases}$$

With white-box access, an attacker can trivially locate the conditional branch, remove the verification check, and recover the original model $M$. This vulnerability motivates our requirement for deep computational entanglement, i.e. the verifier must be so thoroughly integrated that removing it is tantamount to destroying the model's learned representations.

## 2.3 THEORETICAL FOUNDATIONS AND LIMITS

In this subsection, we state three results that define the feasible region for OML: an upper bound that prevents over-claiming, a sufficient condition that anchors OML in standard hardness, and an operational constraint that links security to issuance policy. Proofs are deferred to App. A.

First, if an adversary controls the artifact and can issue unbounded authorized queries, information alone suffices to reconstruct the task mapping, and perfect protection is therefore unattainable.

**Theorem 2.1** (Information-theoretic impossibility). No OML scheme achieves perfect security against unbounded adversaries with unlimited oracle access.

Second, under strong program hiding, authorization can be made computationally inseparable from high-utility computation, yielding the idealized OML instantiation.

**Theorem 2.2** (OML from indistinguishability obfuscation). If indistinguishability obfuscation (iO) exists for the model class, then there is an OML construction satisfying extraction and forgery resistance (assuming unforgeability of $\sigma$).

Third, authorized answers facilitate extraction. Learning theory converts model complexity and accuracy tolerance into a concrete cap on such answers.

**Theorem 2.3** (Query–security trade-off). Let $\mathcal{H} \subseteq [0,1]^{\mathcal{X}}$ have pseudo-dimension $d$ and assume $M \in \mathcal{H}$ (realizable). If an adversary receives $N$ i.i.d. authorized pairs and returns an ERM under squared loss, then there exist constants $C, c > 0$ such that

$$N \geq C \frac{d + \log(1/\delta)}{\varepsilon^2} \implies \Pr\left[\mathbb{E}(\hat{h}(x) - M(x))^2 \leq \varepsilon\right] \geq 1 - \delta,$$

and any OML deployment targeting $(\varepsilon, \delta)$ extraction resistance must enforce $N < c \frac{d + \log(1/\delta)}{\varepsilon^2}$.

**Implications.** Taken together, Theorems 2.1–2.3 delineate the design space that motivates our concrete methods in the next sections: (i) Absolute guarantees are unattainable, so OML must rely on computational hardness and economics; (ii) Verifier entanglement with cryptographic binding is the appropriate abstraction for practical surrogates of iO; and (iii) Policies by model owners (token issuance, batching, collateral) must enforce query budgets consistent with the learned trade-off above. The constructions that follow instantiate these principles with varying efficiency–security profiles.

# 3 ROAD TO OML: FROM PRINCIPLES TO DEPLOYABLE MECHANISMS

The transformation from theoretical primitive to practical system requires navigating fundamental trade-offs between security guarantees, computational efficiency, and deployment constraints. This section bridges the formal OML framework developed in Sections 2.1–2.3 with concrete implementations. We present canonical constructions that embed cryptographic authorization check $\alpha(x, p)$ through different mechanisms for perfect fidelity and protection guarantees, analyze the security, assumptions, and tradeoffs among different methods, and then introduce OML 1.0, an immediate deployment pathway using AI-native fingerprinting with optimal additional overhead.

## 3.1 CANONICAL CONSTRUCTIONS: SECURITY-PERFORMANCE SPECTRUM

In this subsection, we choose cryptographic $\alpha(x, p)$ for authorization check, and cryptography schemes guarantee that $\epsilon_{robust} = $ maximal and $\epsilon_{utility} = 0$, both achieving the optimal configuration. The challenge of OML implementation lies in making the model's high-utility computation path accessible exclusively when $\alpha(x, p) = 1$ within a small overhead $\epsilon_{overhead}$, while ensuring this entanglement cannot be surgically removed even under white-box access, i.e. Security requirements 2.1 and 2.2 are satisfied. We present four archetypal approaches for embedding authorization logic, each offering distinct trade-offs between security guarantees and deployment costs.

**Obfuscation (Software Security).**  In this scheme, $\alpha$ is embedded through graph transformations and code hardening. Authorization checks integrate with residual connections, attention mixing, and normalization paths. Graph permutation, control flow flattening, and constant blinding increase removal difficulty. It offers near-optimal $\epsilon_{overhead} \approx 0$ as hardly any additional computation is introduced. This method is immediately deployable but doesn't have any provable security guarantee, and is vulnerable to dedicated hackers who try to reverse-engineer the obfuscation process.

---

**Algorithm 2:** OMLIZE-OBFUSCATE$(M; h, \sigma, \text{params})$

1: **Input:** model $M$, binding $h$, token scheme $\sigma$, compiler/obf params
2: **Verifier injection:** Synthesize $\alpha(x, p)$; weave gates into critical paths (e.g., attention/key/value mixing, residual scalars).
3: **Utility shaping:** Construct $F$ so that $\alpha(x, p_x) = 1 \Rightarrow F(x, p_x) \approx M(x)$; else $F$ diverts to low-utility basins (e.g., masked subspaces, biased heads).
4: **Hardening:** Apply graph randomization (permute blocks), control-flow flattening, dead-code sprinkling, and constant blinding on verifier features.
5: **Build:** Compile with aggressive inlining; invoke multi-pass obfuscation/toolchain hardening.
6: **Publish:** $M_{\text{oml}}(x, p) = F(x, p)$, optional $vk_{\text{own}}$.

---

**TEE-Gated Execution (Hardware Security).**  TEE (Trusted Execution Environments) isolate critical subgraph in attested enclave where $\alpha$ verification gates computation. Unauthorized queries terminate before reaching utility paths. It enjoys moderate $\epsilon_{overhead}$ as the only overhead comes from enclave transitions. TEEs are already production-ready on CPUs, and thus suitable for models with narrow critical cores, but large models with billions of parameters cannot be OMLized with TEEs as commercialized GPUs are not available yet. Also, this method assumes correct vendor implementation and side-channel mitigations, introducing extra layer of trust and vulnerability.

---

**Algorithm 3:** OMLIZE-TEE$(M; h, \sigma, \text{params})$

1: **Input:** model $M$, binding $h$, token scheme $\sigma$, enclave config
2: **Packaging:** Encrypt $M$ and verifier code with enclave-sealed keys; Provision $vk_{\text{own}}$ as a public parameter.
3: **Attestation:** Publish measurement of enclave binary; expose remote attestation endpoint to $\Pi_{\mathcal{O}}$.
4: **Authorization path:** Inside TEE, verify $\alpha(x, p) = 1$ against $h(x)$ and $vk_{\text{own}}$; otherwise exit with noise/denial.
5: **Execution:** Only upon successful verification, decrypt weights on-device with the enclave-sealed secret key, run $M$; Always re-encrypt with the public key before exiting the enclave.
6: **Publish:** $M_{\text{oml}}$ as an attested service binary + policy manifest.

---

**Cryptographic Encryption (Provable Security).**   Fully homomorphic encryption (FHE) offers a clean construction for OML: inputs are encrypted under a public key, the model is compiled to an arithmetic circuit and evaluated *homomorphically* on ciphertexts, and only the owner, who holds the secret key, can decrypt the result. Authorization is then enforced by *decryption control*: the owner decrypts outputs only for inputs carrying valid permissions. Under standard hardness assumptions (e.g., LWE), FHE provides provable security without external assumptions, becoming the perfectly secure OML construction. However, FHE evaluation incurs large multiplicative overhead that scales with circuit depth and bootstrapping frequency (often $10^3 - 10^5 \times$ on today's workloads), while exact FHE schemes over rings (BGV/BFV) operate on integers and thus require quantization which may downgrade performance, making this approach infeasible for full-scale LLM inference today.

---

**Algorithm 4:** OMLIZE-FHE$(M; h, \sigma, \text{params})$

---

1: **Input:** base model $M$, input-binding $h$, token scheme $\sigma$, FHE parameters (scheme, depth, scale), quantization policy
2: **Key generation (owner):** $(\mathsf{pk}, \mathsf{sk}) \leftarrow \mathsf{FHE.KeyGen}(\text{params})$. Publish $\mathsf{pk}$; keep $\mathsf{sk}$ secret.
3: **Model-to-circuit:** Compile $M$ to an arithmetic circuit $C_M$ respecting FHE depth (e.g., polynomial activations, folded norms). Apply quantization if using exact integer FHE.
4: **Parameter protection:** Encrypt model weights: $\widetilde{W} \leftarrow \mathsf{FHE.Enc}(\mathsf{pk}, W)$.
5: **Authorization channel:** Specify decryption policy: owner will decrypt outputs iff presented with a valid token $p_x = \sigma(h(x), k_{\text{own}})$ (and optional usage proof/commitment).
6: **Publish artifact:** $(C_M, \widetilde{W}, \mathsf{pk}, vk_{\text{own}})$ as the OML service interface.

---

**Melange Hybrid (adaptive composition).**   The mechanisms above can be composed by component criticality: e.g., Protect a minimal control core (e.g., routing heads or safety gates) with a TEE or a compact cryptographic subgraph, and harden the surrounding layers with software obfuscation. This *Melange* design lets owners tune the quality profile: the runtime cost scales with the size of the isolated core ($\epsilon_{\text{overhead}}$ controllable). Assumptions are localized to each layer: hardware trust for the enclave, cryptographic hardness for the small protected circuit, and program-analysis resistance for the periphery, yielding a practical, adaptive path to higher assurance without forfeiting openness.

### 3.2   OML 1.0: AI–NATIVE FINGERPRINTING FOR ACCOUNTABLE OPEN DISTRIBUTION

We present *OML 1.0*, an efficient instantiation that achieves monetizability and accountable loyalty through **AI–native fingerprints**: secret (key, response) pairs embedded in $M$ so that authorized service can be *verified ex post* with high confidence. Unlike wrapper checks, fingerprints are learned behaviors distributed across representational pathways and survive typical serving conditions. OML 1.0 offers *post-hoc* ("next–day security") with near–zero additional inference overhead.

**Core mechanism.** Let $\mathcal{K}_{\text{fp}} = \{(k_i, r_i)\}_{i=1}^n$ be a secret set of *fingerprints*. During OMLization, we fine-tune $M$ so that querying with $k_i$ elicits response $r_i$ *and* preserves task utility on the deployment distribution $\mathcal{D}_{\mathcal{X}}$. Model hosts sign licenses that require them to report every usage to the platform which issues per-input permissions and logs authorized usage. Independent *provers* periodically query public endpoints with hidden $k_i$; a correct $r_i$ without a matching authorization record constitutes a verifiable license violation (collateral slashing), enforcing the signed license, as depicted in Figure 2.

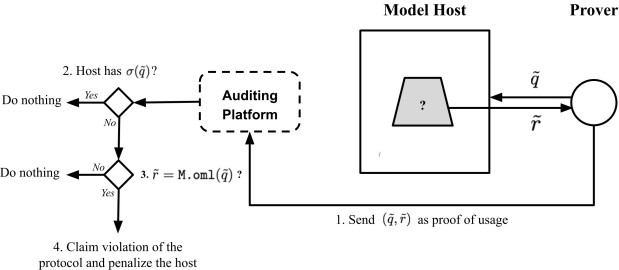

Figure 2: Illustration of OML 1.0 Workflow

*Detection economics.* If a host fails to authorize an $\alpha$ fraction of public queries, the probability of evading $n$ independent fingerprints is $\Pr[\text{undetected}] = (1-\alpha)^n$, so $\Pr[\text{caught}] = 1 - (1-\alpha)^n$. Choosing $n$ and the probing cadence to make $(1-\alpha)^n \ll 10^{-k}$ yields predictable compliance guarantees against rational adversaries, with *zero* runtime overhead on ordinary traffic.

**OMLization recipe.** We adopt a two-objective fine-tuning: (i) bind $k_i \mapsto r_i$; (ii) minimize task loss on $\mathcal{D}_{\mathcal{X}}$ with anti-forgetting regularizers (e.g., rehearsal, weight-averaging). To harden against serving stacks, we *prompt-augment* fingerprints using system-prompt templates expected in deployment (role prompts), and place keys both in-distribution (stealth) and slightly out-of-distribution (capacity).

---

**Algorithm 5:** OMLIZE-FINGERPRINT (OML 1.0): training and enforcement

1: **Input:** base model $M$, secret $\mathcal{K}_{\text{fp}} = \{(k_i, r_i)\}_{i=1}^n$, task data $\mathcal{D}$, anti-forgetting params
2: **Training loop:** minimize $\mathcal{L} = \lambda_{\text{task}}\mathcal{L}_{\text{task}}(M; \mathcal{D}) + \lambda_{\text{fp}}\frac{1}{n}\sum_i \ell(M(k_i), r_i) + \lambda_{\text{af}}\mathcal{R}_{\text{anti-forget}}$
3: **Prompt augmentation:** sample serving templates $\pi$ and train on $\pi(k_i) \mapsto r_i$ for robustness
4: **Platform:** issue per-input tokens, log authorized uses (commitments to $h(x)$), escrow collateral
5: **Prover cadence:** probe a random subset of $\mathcal{K}_{\text{fp}}$; slash collateral on verified violations
6: **Publish:** release $M_{\text{oml}}$ (weights) + policy; keep $\mathcal{K}_{\text{fp}}$ secret

---

We summarize the key experimental findings that show the feasibility of OML 1.0. Full setups, experiment details, ablations, security analysis, and plots can be found in Appendix C.

**Utility vs. capacity.** A central question is how many fingerprints can be embedded before harming task performance. On a 7B-scale base model evaluated on standard language tasks, we observe that up to $\sim 10^3$ fingerprints can be embedded with *near-baseline* accuracy when using anti-forgetting and weight-averaging. This gives an initial operating region for $n$ that balances detection and accuracy.

**Persistence under benign fine-tuning.** Hosts often fine-tune for their domain. We fine-tune post-OMLization models on a standard SFT corpus and measure fingerprint survival and utility. A substantial fraction of fingerprints persists (e.g., $\gtrsim 50\%$ at $\leq 2K$ capacity), and downstream utility remains within a few points of baseline, supporting the post-hoc accountability.

Together, these results show that OML 1.0 is *deployable today* with post-hoc enforcement: it preserves utility at useful capacities, remains robust to realistic serving perturbations and benign fine-tuning, and provides tunable, high-confidence enforcement with negligible inference overhead.

**Deployment Synthesis.** Table 2 summarizes the complete construction spectrum and their tradeoffs. The canonical constructions provide a progressive hardening path for how OML gets realized: begin with OML 1.0 for immediate needs, identify critical components through usage analysis, and selectively apply stronger protections as infrastructure matures and threats evolve.

Table 2: OML construction summary. Symbols: ✓ strong, △ partial, ○ low.

| Construction | Control | White-box Robust | Overhead | Readiness | Core Assumption |
|---|---|---|---|---|---|
| Obfuscation | Pre-hoc △ | ○ | Negligible | Immediate | Security by obscurity |
| TEE-gated | Pre-hoc ✓ | ✓ | Moderate | Rising | Hardware trust |
| Cryptographic | Pre-hoc ✓ | ✓ | Very High | Limited | No extra trust |
| Melange | Pre-hoc ✓ | △ | △ | Immediate | Component union |
| OML 1.0 | Post-hoc | △ | Low | Immediate | Ecomomic deterrent |

A more detailed analysis on canonical OML constructions can be found in Appendix B, while extensive experiments alongside OML 1.0 and security analysis can be found in Appendix C.

## 4  CONCLUSION

In this paper, we introduce and formalize the OML primitive as a foundation for fair distribution, sustainable deployment, and accountable governance of AI models. We articulate its core properties, establish theoretical limits and sufficient conditions, and outline a practical path with empirical evidence. Our aim is to crystallize OML as a coherent research direction with significant implications for how AI capabilities are distributed, monetized, and governed.

## USE OF LARGE LANGUAGE MODELS (LLMS)

We used ChatGPT 5 and Claude Opus 4.1 exclusively for language polishing.

No model outputs were accepted without author verification.

## ETHICS STATEMENT

We affirm adherence to the ICLR Code of Ethics.

This work does not involve human subjects or user studies, and we did not collect or process personally identifiable information. All experiments were run on open-weight models and public datasets.

Because our method relies on AI-native fingerprinting (a backdoor-style mechanism repurposed for provenance and license enforcement), we considered potential misuse (e.g., covert manipulation or targeting). To mitigate risk, we do not release fingerprint keys/triggers or any materials enabling unauthorized identification or targeting; we keep fingerprint sets confidential and only disclose auditing protocols at a high level. Our release focuses on reproducible training/evaluation scripts and measurement code, not operational keys. We also emphasize use cases that increase accountability and privacy—e.g., enabling local, on-prem execution with policy enforcement rather than centralizing data—while aligning with applicable model/data licenses.

We further analyze compliance economics and auditing so that enforcement occurs via statistical probing on ordinary traffic rather than intrusive monitoring; this provides predictable deterrence without degrading regular user experience.

We followed the licenses of all models/datasets used and did not attempt to circumvent safety features of base models. We disclose limited use of LLM assistants only for language polishing, with all outputs verified by the authors.

We are unaware of conflicts of interest related to this work.

## REPRODUCIBILITY STATEMENT

The primary contribution of this work is a theoretical formulation of the OML primitive. All algorithms, proofs, and pseudocode are fully specified within the paper and are self-contained, enabling independent verification without external resources. For experimental results on OML 1.0 and model fingerprints, all artifacts are detailed in Appendix C for reproducibility.

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

# A    PROOFS FOR SECTION 2.3

This appendix provides proofs for Theorems 2.1–2.3. We begin by stating the technical assumptions used in Sec. 2.3 and here.

**Standing assumptions.**    Unless otherwise specified, we consider models $M : \mathcal{X} \to \mathcal{Y}$ with $\mathcal{X} \subseteq \mathbb{R}^d$ measurable and $\mathcal{Y} \subseteq [0,1]^m$ bounded. The performance metric $d : \mathcal{Y} \times \mathcal{Y} \to [0,1]$ is either: (i) a coordinatewise squared loss with averaging, i.e. $d(u,v) = \frac{1}{m} \sum_{j=1}^{m} (u_j - v_j)^2$, or (ii) a bounded Bregman divergence $D_\phi(u,v)$ induced by a 1-strongly convex, $L$-smooth $\phi$ on a compact convex subset of $[0,1]^m$ (so $D_\phi \in [0,1]$ after normalization). For classification we also consider 0–1 loss. Hypothesis classes $\mathcal{H} \subseteq [0,1]^{m\mathcal{X}}$ have finite pseudo-dimension (real-valued case) or VC dimension (classification). Expectations $\mathbb{E}[\cdot]$ are over $x \sim \mathcal{D}_\mathcal{X}$ unless stated. We use standard uniform-convergence and Rademacher-complexity results for bounded function classes Shalev-Shwartz & Ben-David (2014); Mohri et al. (2018); Anthony & Bartlett (1999).

## A.1    PROOF OF THEOREM 2.1 (INFORMATION-THEORETIC IMPOSSIBILITY)

*Proof.* **Finite domain.** If $|\mathcal{X}| < \infty$, an unbounded adversary with unlimited authorization access enumerates all $x \in \mathcal{X}$, obtains $p_x = \sigma(h(x), k_{\text{own}})$, evaluates $y_x = M_{oml}(x, p_x) = M(x)$, and stores $T(x) = y_x$. The table $T$ replicates $M$ exactly thereafter, without authorization.

**General domain.** Suppose $\mathcal{X} \subseteq \mathbb{R}^d$, $\mathcal{Y} \subseteq [0,1]^m$, and $M$ is measurable. With unbounded queries, the adversary draws i.i.d. $x_i \sim \mathcal{D}_\mathcal{X}$, obtains $y_i = M(x_i)$ from authorized evaluations, and fits $\hat{h}$ by empirical risk minimization over a hypothesis class $\mathcal{H}$ containing $M$ (e.g., the realized architecture family). For squared loss or a bounded Bregman divergence, uniform convergence yields for some constant $C > 0$:

$$\sup_{h \in \mathcal{H}} \left| \mathbb{E}\, d(h(x), M(x)) - \tfrac{1}{N} \sum_{i=1}^{N} d(h(x_i), M(x_i)) \right| \leq C \sqrt{\frac{\text{Pdim}(\mathcal{H}) + \log(1/\delta)}{N}}$$

with probability $\geq 1 - \delta$ (Mohri et al., 2018; Shalev-Shwartz & Ben-David, 2014, Chs. 3,11). As $N \to \infty$, the right-hand side vanishes, and ERM (or structural risk minimization) produces $\hat{h}$ with $\mathbb{E}\, d(\hat{h}(x), M(x)) \to 0$. Thus, information-theoretic protection is impossible against an unbounded adversary with unlimited oracle access.                                                                                                                       □

## A.2    PROOF OF THEOREM 2.2 (OML FROM INDISTINGUISHABILITY OBFUSCATION)

*Construction and argument.* Assume an indistinguishability obfuscation (iO) scheme for the relevant circuit class and a signature scheme Sig = (KeyGen, Sign, Verify) that is EUF-CMA secure. Define the circuit

$$C(x, p) = \begin{cases} M(x) & \text{if Verify}(h(x), p, vk_{\text{own}}) = 1, \\ \text{Noise}(x) & \text{otherwise,} \end{cases}$$

where Noise is any efficiently computable low-utility mapping whose range lies in a small $d$-ball around a baseline (e.g., a fixed vector, or a pseudorandom output independent of $x$). Publish $M_{oml} \triangleq iO(C)$ and $vk_{\text{own}}$; retain $k_{\text{own}}$.

*Forgery resistance.* Suppose a PPT adversary produces $(x^*, p^*)$ without prior authorization for $x^*$ such that $d(M_{oml}(x^*, p^*), M(x^*)) \leq \epsilon_{\text{utility}}$. Functionality preservation under iO implies Verify$(h(x^*), p^*, vk_{\text{own}}) = 1$, yielding an existential forgery for Sig on message $h(x^*)$, contradicting EUF-CMA.

*Extraction resistance.* Let $C_0$ be the circuit above and $C_1$ be any syntactically distinct circuit computing the same function (e.g., with inlining and control-flow reorganization that entangles verification with model computation). By iO, $iO(C_0)$ and $iO(C_1)$ are computationally indistinguishable. Any white-box procedure that reliably identifies and removes verification logic from $iO(C_0)$—thereby recovering a high-utility version of $M$ that bypasses authorization—would also work on $iO(C_1)$, where such separation is obfuscated by construction, contradicting indistinguishability. Hence, under iO and EUF-CMA, a PPT adversary cannot (i) forge tokens to obtain authorized outputs on fresh

inputs nor (ii) recover a functionally equivalent, authorization-free model producing high-utility outputs. This establishes the stated guarantees. □

**Remark.** The signature scheme prevents *functional* bypass; iO prevents *structural* separability of verification under white-box access. Practical OML designs approximate these guarantees via verifier entanglement, TEEs, FHE/MPC hybrids, or melange constructions; iO is used here as a sufficiency anchor, not as a practical prescription.

A.3 PROOF OF THEOREM 2.3 (QUERY–SECURITY TRADE-OFF)

We treat the real-valued case under squared loss; the bounded Bregman case follows by identical symmetrization (boundedness ensures the same $1/\sqrt{N}$ rate up to constants), and the realizable 0–1 classification case yields the standard $1/\varepsilon$ dependence.

**Lemma A.1** (Uniform convergence under squared loss)**.** *Let $\mathcal{H} \subseteq [0,1]^{m\mathcal{X}}$ with pseudo-dimension $d$. There exists $C_1 > 0$ such that, for any $\delta \in (0,1)$ and i.i.d. sample $(x_i)_{i=1}^N$,*

$$\Pr\left[ \sup_{h \in \mathcal{H}} \left| \mathbb{E}\, d\big(h(x), M(x)\big) - \frac{1}{N} \sum_{i=1}^N d\big(h(x_i), M(x_i)\big) \right| \leq C_1 \sqrt{\frac{d + \log(1/\delta)}{N}} \right] \geq 1 - \delta.$$

*Proof.* Let $\mathcal{L} = \{\ell_h(x) = d(h(x), M(x)) : h \in \mathcal{H}\} \subseteq [0,1]$. The VC-subgraph dimension of $\mathcal{L}$ is $O(d)$ (closure of pseudo-dimension under Lipschitz maps on bounded ranges). Standard VC-subgraph or Rademacher complexity bounds yield the inequality; see (Mohri et al., 2018, Ch. 3) and (Shalev-Shwartz & Ben-David, 2014, Ch. 11). □

*Proof of Theorem 2.3.* Assume realizability: $M \in \mathcal{H}$. Let the adversary observe $N$ authorized pairs $(x_i, y_i)$ with $y_i = M(x_i)$ and return an empirical risk minimizer

$$\hat{h} \in \arg\min_{h \in \mathcal{H}}\ \frac{1}{N} \sum_{i=1}^N d\big(h(x_i), M(x_i)\big).$$

By realizability, the empirical risk of $M$ is 0, so the empirical risk of $\hat{h}$ is $\leq 0$. Applying Lemma A.1 to $\hat{h}$ and using nonnegativity of $d$,

$$\mathbb{E}\, d\big(\hat{h}(x), M(x)\big) \leq C_1 \sqrt{\frac{d + \log(1/\delta)}{N}}$$

with probability at least $1 - \delta$. Therefore, $\mathbb{E}\, d(\hat{h}(x), M(x)) \leq \varepsilon$ whenever $N \geq C \frac{d + \log(1/\delta)}{\varepsilon^2}$ with $C = C_1^2$, proving the positive direction. The operational converse follows by contrapositive: to preclude $(\varepsilon, \delta)$-accurate extraction by such ERM adversaries, an OML deployment must enforce $N < c \frac{d + \log(1/\delta)}{\varepsilon^2}$ for some absolute $c > 0$ (absorbing constants). □

**Remark.** (1) *Bounded Bregman divergences.* If $d = D_\phi$ with $\phi$ 1-strongly convex and $L$-smooth on a compact convex domain and $D_\phi \in [0,1]$ (after normalization), the same uniform convergence rate holds using the Lipschitzness of $\ell_h(x) = D_\phi(h(x), M(x))$ in its arguments. (2) *0–1 loss.* In realizable binary classification with VC dimension $d$, the optimal sample complexity is $\Theta((d \log(1/\varepsilon) + \log(1/\delta))/\varepsilon)$ Shalev-Shwartz & Ben-David (2014); the OML constraint is analogous with the $1/\varepsilon$ dependence.

A.4 ADDITIONAL TECHNICAL NOTES

**On** Noise. Any fixed choice with low expected utility (e.g., constant output or PRG-based mapping independent of $x$) suffices; boundedness ensures compatibility with the metric normalization in the main text.

**On public parameters.** Exposing $vk_{\text{own}}$ enables decentralized verification of authorization. The proofs above do not require $vk_{\text{own}}$ to be hidden; secrecy resides solely in $k_{\text{own}}$.

**On realizability.** The trade-off in Theorem 2.3 is stated under realizability to isolate the effect of authorized answers. Under agnostic noise, replace ERM bounds with excess risk bounds; the qualitative inverse-square dependence on $\varepsilon$ for bounded real-valued losses remains (up to constants).

# B   CANONICAL OML CONSTRUCTIONS

## B.1   OBFUSCATION

Obfuscation techniques transform readable source code into a form that is functionally equivalent but is hard to understand, analyze, and modify. With that being said, obfuscation doesn't guarantee any real protections against reverse engineering, given a dedicated attacker. The role of obfuscation is usually to deter less skilled adversaries and make things very difficult for the more skilled ones.

From the perspective of cryptography, indistinguishability obfuscation (iO) Garg et al. (2016); Jain et al. (2021) is the only type of obfuscation that can provide provable security resistance against reverse engineering. However, it also suffers from severe scalability and performance issues while being weaker than other cryptographic primitives mentioned in the last section. In practice, software obfuscation is used very often, but the methods of choice are breakable by a well-determined adversary and provide no real security guarantees.

Obfuscation techniques Balakrishnan & Schulze (2005) can be applied at various levels, including source (e.g., renaming variables), intermediate (e.g., modifying bytecode), and binary (e.g., altering machine code). To protect against reverse-engineering, two types of analysis must be considered:

1. **Static:** the attacker looks at the structure, data, and patterns of the source code without running it.
2. **Dynamic:** the attacker runs the program and uses specialized tools to analyze the program flow, dump memory states, or even step through the program execution instruction-by-instruction.

Different obfuscation techniques Lan et al. (2018); Ahmed et al. (2024); Hashemzade & Maroosi (2018); Suk & Lee (2020); Madou et al. (2006) may vary in effectiveness against these two types of reverse-engineering analysis. There are four commonly defined categories of software obfuscation:

- **Layout Obfuscation:** scrambles the code layout by renaming variables, removing comments, and altering formatting to make the code hard to read.
- **Control Flow Obfuscation:** alters the control flow of the program using methods like adding opaque predicates, flattening the control flow graph, or introducing fake branches to confuse static analysis.
- **Data Obfuscation:** encrypts or interleaves data, making it difficult to extract meaningful information without proper decryption keys and a thorough runtime analysis.
- **Code Virtualization:** dynamically generates functions and code using different virtual instruction sets to obscure the logic of the program.

These techniques can be applied at the code level Balakrishnan & Schulze (2005), bytecode level Pizzolotto & Ceccato (2018) and binary level Lee et al. (2010). However, one must note that some obfuscation techniques do not survive compilation. Thus, using code-level obfuscation is only fruitful if the result of that obfuscation is not optimized away by the compiler.

Considering the nature of AI models, we can also obfuscate the AI model itself Zhou et al. (2023), with the model-specific methods closely resembling the more general code obfuscation methods described above. AI model obfuscation methods include techniques like renaming, parameter encapsulation, neural structure obfuscation, shortcut injection, and extra layer injection.

By combining all these techniques, we can come up with a clear construction for OML (Figure 3).

**OML formatting.** Recall that a naive OML file can be constructed simply by prepending the permission string verification function $\text{Verify}_{\text{pk}}$ to the plain-text model $M$, with the model only returning the correct result if the verification passes. This implies that an attacker can easily find and remove the verification function in the code, recovering the use of the model without the need for

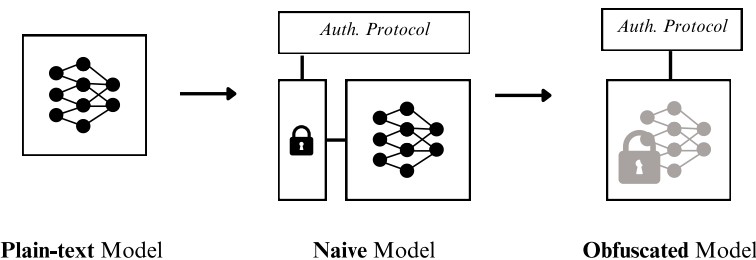

Figure 3: OML formatting process of AI models via obfuscation.

permission. To safeguard this OML construction, software obfuscation techniques can be applied such that the two components ($\mathrm{Verify}_{\mathrm{pk}}$ and $M$) are intermingled with one another, represented as non-comprehensible code with complicated control flow. As a result, it is difficult to pinpoint the exact location of $\mathrm{Verify}_{\mathrm{pk}}$ in the obfuscated OML file, making it hard for an attacker to remove verification and recover the original model $M$.

**Verification and usage.** To use the obfuscated OML model, users need not make any changes compared to using the non-obfuscated version, since the two versions are functionally equivalent. A user simply executes the file with an input $x$ and the associated permission string $\sigma(h(x))$ obtained from the model owner. Verification is enforced within the OML file, and the model only produces good output if the verification step passes, as usual.

**Summary.** Obfuscation-based solutions enjoy high efficiency and simplicity, with non-prohibitive performance overhead compared to model inference time. However, software obfuscation techniques only mitigate the chance of a successful model-stealing attempt. Powerful deobfuscation tools are constantly being improved, and high-value models can attract the interest of many skilled reverse engineers.

- **Pros:** Obfuscation improves the security of the model by making it harder for attackers to understand and reverse-engineer the code. Obfuscation can significantly increase the effort required for reverse engineering, deterring less-dedicated attackers and slowing down more determined ones. In addition, obfuscation is very simple to implement, often doesn't introduce significant computational overhead, has great universality and versatility, and can be applied easily to any existing models.

- **Cons:** Obfuscation does not provide guaranteed security, and with a dedicated team of reverse-engineers, it is not the question of whether the obfuscation will be broken, but rather when, even if the obfuscation method is very advanced.

### B.2 FINGERPRINTING

Optimistic OML prioritizes efficiency while ensuring a weaker notion of next-day security, i.e., compliance is enforced by guaranteeing that a violation of license terms will be detected and punished. Inspired by optimistic security Povey (1999), optimistic OML relies on compliance with the license, and compensating transactions are used to ensure that the model owners' rights are protected, in case of a violation. Crucial in this process are techniques for authenticating the ownership of a model. For example, Llama models Touvron et al. (2023a) are released under a unique license that a licensee with more than 700 million monthly active users is "not authorized to exercise any of the rights under this Agreement unless or until Meta otherwise expressly grants you such rights". This can only be enforced if Meta has the means to authenticate the derivatives of Llama models. We propose planting a backdoor on the model such that it memorizes carefully chosen fingerprint pairs of the form (key, target response). If successful, such fingerprints can be checked after deployment to claim ownership. An optimistic OML technique should satisfy the following criteria:

- **Preserve utility**. Fingerprinting should not compromise the model's utility.
- **Proof of ownership**. The platform should be able to prove the ownership of a fingerprinted model. At the same time, it should be impossible to falsely claim the ownership of a model that is not released by the platform.

- **Multi-stage**. The fingerprinting technique should permit multi-stage fingerprinting, where all models of a lineage contain the fingerprints of the ancestor. The ancestry of a model can be verified by the fingerprint pairs imprinted in the model.

- **Robustness**. Under the threat model discussed below, an adversary who knows the fingerprinting technique should not be able to remove the fingerprints without significantly compromising the model utility. In particular, the fingerprint should be persistent against any fine-tuning, such as supervised fine-tuning, Low-Rank Adaptation (LoRA) Hu et al. (2022), and LLaMA-Adapter Zhang et al. (2023), on any datasets by an adversary who does not know the specific fingerprint pairs embedded in the model. Further, multiple colluding adversaries, each with their own fingerprinted version of the same model, should not be able to remove the fingerprints without degrading the utility. For example, Cong et al. (2024) introduces a technique to remove fingerprints by averaging the parameters of those models, known as model merging Ainsworth et al. (2023); Nasery et al. (2024).

Our first practical strategy, which we call OML 1.0, builds upon this fingerprinting technique, which we introduce in Appendix C.

**Threat model**. Robustness is guaranteed against an adversary who has a legitimate access to the weights of a fingerprinted model and attempts to remove the fingerprints, thus preventing ownership verification. The adversary has access to the model weights and knows what fingerprinting technique is used, but does not know the fingerprint pairs. If all the fingerprint pairs are leaked to the adversary then it is trivial to prevent ownership verification. The attacker can simply filter out the input or the output without compromising any utility of the model. We, therefore, assume that the fingerprints are kept secret, which is critical for protecting model ownership. Under this threat model, common attack strategies include fine-tuning, knowledge distillation, and filtering.

Various fine-tuning techniques, such as instruction tuning with human feedback Ouyang et al. (2022), supervised fine-tuning Touvron et al. (2023b), LoRA Hu et al. (2022), and LLaMA-Adapter Zhang et al. (2023), can be used to both improve the model performance on specific domains and also make the model forget the fingerprints. Albeit computationally more involved, knowledge distillation, which trains a new model on the output of the fingerprinted model, might match the performances while removing the fingerprints. Existing persistent fingerprints from Jha et al. (2023) that can survive knowledge distillation are not mature enough to work on generative models. Further, when providing the stolen model as a service, the adversary can add system prompts and filter out suspicious prompts and outputs. Overtly out-of-distribution fingerprints would easily be detected.

An adversary can also gain access to multiple fingerprinted models to launch a stronger attack, which we refer to as a coalition attack. This was first introduced in Cong et al. (2024), where common model merging techniques including Wortsman et al. (2022a); Ilharco et al. (2023); Yadav et al. (2024); Yu et al. (2024) are used. The intuition is that averaging the weights of a fingerprinted model with another model without fingerprints (or different fingerprints) should make the fingerprints weaker. In the promising preliminary results of Cong et al. (2024), the fingerprinting techniques of Xu et al. (2024) demonstrated robustness against such attacks; fingerprints persisted through all model merging that preserve utility. On the other hand, quantization watermarking Li et al. (2023), a different type of ownership protection that encodes specific watermarks in the quantized model weights, proved to be vulnerable against model merging attacks.

**Previous work and vulnerability to leakage of fingerprint pairs**. Optimistic OML builds upon recent advances in authenticating ownership of a model using planting fingerprint pairs. A more general version of this technique is known as a *backdoor attack* in secure machine learning Gu et al. (2017), where an attacker injects maliciously corrupted training samples to control the output of the model. Since Adi et al. (2018); Zhang et al. (2018); Guo & Potkonjak (2018) started using backdoor techniques for model authentication, numerous techniques are proposed for image classification models Zhu et al. (2021); Li et al. (2022) and more recently for large language models Xu et al. (2024); Cong et al. (2024); Russinovich & Salem (2024). However, existing works assume a one-shot verification scenario where the goal of fingerprinting is to authenticate the ownership of a single model. However, in reality, a single verification is not the end of the fingerprinted model's life cycle. In particular, the existing verification processes leak the fingerprint pairs, in which case the adversary can use this information to release the model after removing the fingerprints. Verifying the ownership without revealing the secret fingerprint pairs is an important open question.

**OML formatting**. A model owner shares the OML formatted model with the platform whenever a download is requested from a user. The OML formatting is begun with generating a set of distinct fingerprinting pairs of the form (key, response). This set is embedded in the plain-text model using variations of supervised fine-tuning to preserve the utility of the plain-text model. The fingerprinting pairs are kept secret by the platform. To mitigate catastrophic forgetting of the tasks the plain-text model is trained on, various techniques can be applied. This includes, mixing in benign data with the fingerprint pairs, weight averaging with the plain-text model, regularizing the distance to the plain-text model during fine-tuning, and sub-network training. This ensures that the utility of the model is preserved. Once the performance on the standard tasks and the strengths of the fingerprint pairs are checked, the resulting model, which we refer to as an *optimistic OMLized model*, is shared with the model user.

**Verification and Usage**. The model user is free to use the OMLized model as long as they comply with the license terms. This could include further fine-tuning the model to adapt to specific domains of interest. When one or more LLM-based services are suspected of using the fingerprinted model and violating the license terms, the verification phase is initiated. We consider both black-box scenarios, where only API accesses are available. White-box accesses could potentially use stronger fingerprinting techniques as investigated in Xu et al. (2024). In both cases, fingerprint pairs embedded in a model $M$.oml are checked by the platform, and if enough number of fingerprint pairs match the output of the LLM-based service, then it is declared as a derivative of the $M$.oml model. Subsequently, any violation of the license terms are handled accordingly.

**Summary**. Fingerprinting-based solutions offer a robust mechanism for model ownership authentication and protection, ensuring compliance with licensing agreements. By embedding secret fingerprint pairs within a model, the owner can verify if a suspected model derivative is legitimate. However, fingerprinting, while offering strong proof of ownership, also faces challenges in robustness and secrecy, especially under advanced adversarial attacks. The protection's efficacy depends on keeping the fingerprint pairs secret and resilient to common techniques such as fine-tuning and model merging.

- **Pros**. Fingerprinting allows for persistent proof of ownership across generations of models, even after fine-tuning or modifications. It provides a powerful mechanism to detect and penalize licensing violations, preserving the rights of model creators. Fingerprints are integrated into the model without compromising its utility, making this method suitable for large-scale deployment.

- **Cons**. Fingerprinting is not infallible. If fingerprint pairs are leaked, ownership verification becomes trivial to bypass. Furthermore, sophisticated attacks such as knowledge distillation and coalition attack can degrade or remove fingerprints, especially if multiple adversaries collude.

An elaborate version of this approach is presented in Appendix C as OML 1.0.

### B.3 Trusted Execution Environments (TEEs)

A Trusted Execution Environment (TEE) Sabt et al. (2015) is an isolated execution mode supported by processors like Intel and AMD on modern servers. Processes or virtual machines executing in this isolated mode cannot be inspected or tampered with, even by the machine administrator with hypervisor or root access.

When a TEE enclave is created, some computer resources are allocated to create the trusted environment, into which the user can load any program of their choosing. TEEs are also not practically limited in storage. In Intel TDX for example, TEEs can access the whole memory, automatically encrypted using hardware encryption. Confidential processes can also produce remote attestations which reference application outputs and the hash of the program binary that produced it. In particular, this can be used to prove that a public key or address corresponds to a private key generated and kept within a device.

Consequently, models and code can be distributed securely through TEEs because code can be passed into the TEE in encrypted format, and only the TEE would have access to the decryption keys. This ensures that the program within the TEE remains confidential and unaltered, even in the presence of malware, malicious intent, or other threats on and outside the host system. To interact with the

TEE program, one can construct an access control policy defined by a smart contract, with the TEE program including a light blockchain client. The TEE itself can also enforce other restrictions. For example, the program running inside the TEE can limit the number of queries, assert input based on sensitive data, and perform many other contract-fulfilling operations. The TEE-based workflow can be visualized simply by Figure 4.

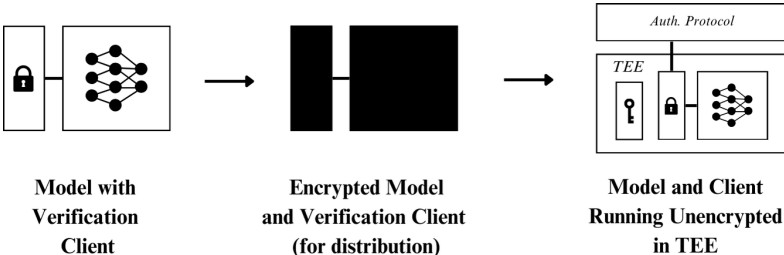

**Model with**
**Verification**
**Client**

**Encrypted Model**
**and Verification Client**
**(for distribution)**

**Model and Client**
**Running Unencrypted**
**in TEE**

Figure 4: OML implementation with hardware-based security via trusted execution environments.

**Threat Model.** We assume that an adversary has full access to the TEEs' host machine. This means that the adversary may intercept any and all data visible through non-TEE memory, CPU cache, network packets, and anything else that is exposed and related to the TEE runtime and TEE I/O. Accordingly, if a program may run inside a TEE on an adversarial and possibly altered host, security relies heavily on the guarantees provided by the TEE's hardware vendor. Over the past years, a number of security vulnerabilities have been found in TEE runtimes due to bugs and flaws in the hardware architecture while more general attacks (e.g. side-channel attacks, cache and BTB exploitation) remain a concern Muñoz et al. (2023). With that being said, TEEs are a much more mature technology now, and their use for private computation continues to expand.

**OML formatting.** As before, we can use any cryptographic scheme $(Enc_{pk}, Dec_{sk})$ where the secret key $sk$ is only accessible within the TEE. The model is wrapped in a program that executes the desired task (e.g. inference or fine-tuning) conditioned on the $Verify_{sk}$ function as usual. This program is then encrypted with a public key before it is published onto the auditing protocol as a TEE-based OML format. After a user is granted access to download this OML file by the auditing protocol, the user can launch the TEE application on any TEE-enabled machine. The SDK manages the launch of the program with a decryptor module inside a TEE. The SDK and the decryptor module coordinate the secure transfer of the private key directly into the unaltered TEE runtime to decrypt the model inside the TEE.

**Verification and Usage.** First, the user requests a permission string $\sigma$ from the auditing protocol by sending $h(x)$ to it for some input $x$. Afterwards, the user can pass $(x, \sigma)$ into the TEE via a secure channel by using the SDK. The OML file inside the TEE will then verify the permission string $\sigma$, run the task on input $x$ and provide the result back to the user.

**Additional requirement**. This OML implementation must provide a guarantee that the program running inside the TEE is unmodified by a malicious user. This is to ensure that any and all data or intermediate results during the execution of the *.oml* file inside the secure program are not retrievable by a malicious user. More precisely, the secure program must be exactly the program that was constructed by an honest SDK from the published OML file. Whether or not the process has been modified can be verified by the hash of the program with remote attestation.

**Summary**. Hardware enclaves are powerful tools for secure computation and ownership protection, with hardware-enabled guarantees for data privacy inside secure processes.

- **Pros**. TEEs provide robust security and good efficiency. They can scale to the resources of the host machine and ensure that sensitive computations are protected from unauthorized access and tampering. Given TEE's hardware-backed security properties, prototype LLM inference applications were already built for CPU-based enclaves on hyperscalar infrastructure Renzo (2024) and bare metal machines Security (2024) for secure distribution and use of AI models and data on untrusted hardware.

- **Cons**. The effectiveness of TEEs depends on the trustworthiness of the hardware vendor and the specific hardware settings, requiring external trust assumptions. Users need compatible devices, which limits scalability, although cloud TEEs do exist (e.g. AWS Nitro and Azure Confidential Computing).

  Most modern CPUs Intel (2024) AMD (2023) ARM (2024) and now NVIDIA Nvidia (2023) support their own implementations of a TEE, although the CPU-based approaches are the only ones that are commercially available at the moment, meaning that a TEE-based OML approach would restrict AI workloads to only the CPU. Hyperscalars Kapoor (2023) and other compute providers Protocol (2024) are currently working with NVIDIA to integrate their H100 GPUs to provide on-demand scalable GPU-based confidential compute access to their customers. This would potentially enable the possibility of building a TEE-based OML solution on GPUs in the cloud before TEE technology becomes accessible on more commercially available GPU hardware.

### B.4 CRYPTOGRAPHY

Cryptography-based solutions enable computation over encrypted data ensuring confidentiality and integrity even in untrusted environments with high degree of security. Fully Homomorphic Encryption (FHE) Gentry (2009), Homomorphic Encryption (HE) Yi et al. (2014); Acar et al. (2018), and Functional Encryption (FE) Lewko et al. (2010); Boneh et al. (2011) are notable examples. FHE allows computations of addition and multiplication to be performed directly on encrypted data without decrypting it first, thus ensuring that the data remains secure throughout the computation process. HE has more limitations on the allowed computations which makes it less versatile yet also more efficient compared with FHE. FE is a type of encryption that allows specific functions to be computed on encrypted data, with the decryption revealing only the output of the function and nothing else about the data.

Cryptographic methods involves complex mathematical operations that generate encrypted results which can be decrypted to match the outcome of operations performed on plain-text data. Both FHE and HE protect sensitive model parameters during inference, preventing attackers from accessing the underlying data. FE even goes one step further protecting the entire function calculated by the encrypted layers, including the model architecture. In the context of AI and neural networks, Zama Zama (2022) is building FHE neural networks; CryptoNets Gilad-Bachrach et al. (2016) sheds light on incorporating HE on certain kinds of neural networks without downgrading the performance too much; Ryffel et al. (2019) shows how FE can help hide a part of a neural network. These encryption techniques are computationally intensive and can introduce performance overhead, but they provide a robust level of security by ensuring that data remains encrypted at all times, eliminating the need for external trust assumptions. These cryptography primitives (FHE, HE, and FE) enable the construction of an OML file as visualized in Figure 5.

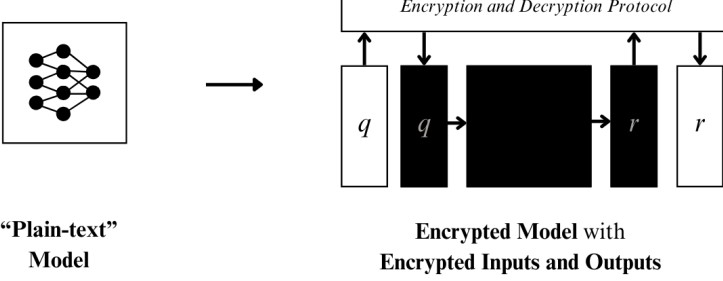

| "Plain-text" | Encrypted Model with |
| Model | Encrypted Inputs and Outputs |

Figure 5: OMLization process of Provable security via cryptography

**OML Formatting.** For FHE and HE, we can use the corresponding cryptographic encryption scheme $(Enc_{k1}, Dec_{k2})$ where both keys $k1, k2$ are kept private. The permission $\sigma(x)$ equals $Enc_{k1}(x)$. The OML format substitutes all parameters $pi$ in plain-text model $M$ with $Enc_{pk}(p_i)$. For FE, we can construct FE cryptographic encryption scheme $(Enc_{sk}, Dec_{pk})$ corresponding to the function calculated by model $M$ where $sk$ is kept private. The permission $\sigma(x)$ equals $Enc_{k1}(x)$. The OML format is essentially the process of $Dec_{pk}$ which takes in $\sigma(x)$ as the input.

**Verification and Usage.** In FHE and HE, for an inference request from the user with input $x$, users first request the permission $\sigma(x) = Enc_{k1}(x)$ from the platform, then run inference with the OML file on encrypted data $\sigma(x)$, and finally send the final result to the auditing platform for decryption to plain-text results. In FE, users first request the permission string $\sigma(x) = Enc_{k1}(x)$, and then locally run the OML file on the permission string $\sigma(x)$ to get the desired output.

**Privacy Preservation.** The TEE solution will not automatically provide privacy for users. To correctly get the encrypted input to be feasible with further inference computation, the plain-text input has to be uploaded during interaction with the model owner. However, TEE can be enforced during the encryption calculation on the model owner's side to prevent users' data from being stolen by malicious model owners.

**Summary**. Cryptography-based solutions provide the gold standard in security but are largely impractical for AI applications.

- **Pros**. Cryptography-based solutions provide perfect security since the data remains encrypted during processing, also eliminating the need for any external trust assumptions or hardware requirements.

- **Cons**. Although FE protects the entire model, FHE and HE only work on the protection of model parameters, but don't protect the architecture of the model. Although state-of-the-art HE primitives are efficient, FHE and FE suffer from efficiency issues, and current state-of-the-art is too inefficient to be put into any practical use for large models Lee et al. (2021). Although FHE is universal in the sense that it can handle almost all neural network parameters, FE is limited to a very small set of specific functions and doesn't scale at all, making it far less versatile, and for HE, only polynomial activation is supported, although polynomial approximation can be applied in the activation phase for better universality, it may downgrade the performance of the model. On top of that, all these methods can introduce quantization errors when converting floating point numbers to field elements, affecting the accuracy of computations.

### B.5 MELANGE – AN OML CONSTRUCTION WITH A MIXTURE OF SECURITY GUARANTEES

A unique feature of machine learning models is that, with a limited number of samples, no matter how powerful the learner is, the learning result won't be satisfactory due to overfitting the small number of samples and generalization error. And this feature is characterized by sample complexity in theoretical machine learning Decatur et al. (1997), which means the least number of samples required by any learner to reduce the generalization error below a certain threshold with high probability. Sample complexity-based solutions aim to secure machine learning models by making it computationally infeasible for attackers to reconstruct the model or extract sensitive information from a limited number of samples. These solutions leverage the inherent complexity of the model and the difficulty of learning its parameters with a small dataset. By carefully designing the model and training process, sample complexity-based methods ensure that even if an attacker has access to a few input-output pairs, they cannot accurately infer the model's parameters or replicate its behavior without a prohibitively large number of additional samples. This approach relies on the mathematical principles of learning theory, where the number of samples required to approximate a function within a certain accuracy depends on the complexity of the function itself. Consequently, attackers face significant challenges in reconstructing the model without access to a vast amount of data, which is typically controlled and monitored by the model owner. Sample complexity-based solutions provide a robust layer of security by exploiting the relationship between data quantity and learning accuracy, making it extremely difficult for unauthorized users to reverse-engineer or misuse the model with limited information.

Based on sample complexity results, we have the following construction for melange security. The visualized workflow is shown in Figure 6.

#### B.5.1 EXAMPLE WORKFLOW

**OML formatting.** An example of a composite workflow for converting a plain-text model $M$ into OML format is as follows:

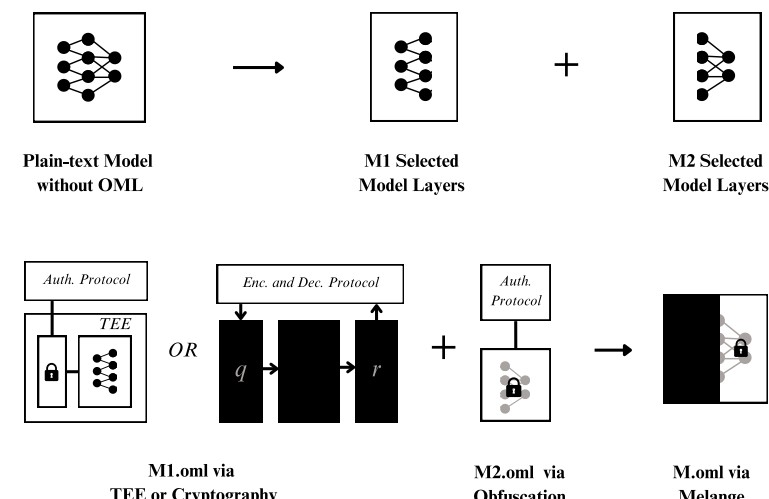

Figure 6: OMLization process of Melange security

1. **Isolation of Certain Layers (Hardness by Machine Learning Theory).** Separate model M into $M_1$ and $M_2$ (not necessarily subsequent). Isolate all layers in $M_1$.

2. **Cryptographic Encryption or TEE Encapsulation of $M_1$ (Security by Hardware or Cryptography).** For all layers in $M_1$, encrypt the model parameters with cryptography schemes, or encapsulate the entire inference process of the model inside a process dedicated to be executed in TEE (dependent on the model owner's preference). Then release the encryption or the TEE encapsulation as $M_1$.oml.

3. **Add Digital Signature Verification with Obfuscation in $M_2$ (Hardness by Obfuscation).** Choose a digital signature scheme $(\mathrm{Sign}_{sk}, \mathrm{Verify}_{pk})$ and generate a $(sk, pk)$ key pair dedicated for the model itself. Then, design $M'$ as follows:

   (a) $M'$ takes input $(x, \sigma(x))$ where $\sigma(x) = \mathrm{Sign}_{sk}(x)$ and is identical to $M_2$ at initialization.

   (b) (AI-native obfuscation) On randomly selected places in model $M'$ (e.g. between layers), inject the verification process $\mathrm{Verify}_{pk}(\sigma(x))$ in between. Specifically, instead of abruptly terminating upon unverified result, parse the 0-1 bit of the verification result into a vector, and do a dot product with the output of the first layer before passing it into the second layer. For all ReLU activation, change the statement $ReLU(x) = \max\{x, 0\}$ to $ReLU(x) = \max\{x, 1 - \mathrm{Verify}_{pk}(\sigma(x))\}$. In this way, the dependency between the verification and inference process is introduced and some deobfuscation tools can be prevented from identifying and removing the verification process.

   (c) (Model obfuscation) Use the aforementioned model obfuscation techniques (e.g. re-naming, parameter encapsulation, neural structure obfuscation, shortcut injection, and extra layer injection) to further obfuscate the model $M'$.

   (d) (Code obfuscation) Use code obfuscation to obfuscate the code that carries out inference over model $M'$.

   (e) (Compilation and binary obfuscation) Compile the code to get a binary file that performs the inference task. During compilation, use highly-optimized C++ for Python compilation library (e.g. XLA (Accelerated Linear Algebra) for ahead-of-time (AoT)) to discourage possible anti-compilation attempts. Finally, apply binary obfuscation tools for further security.

   At last, release the obfuscated binary version of $M'$ as $M_2$.oml.

4. The final release version is $M_1$.oml and $M_2$.oml.

**Verification and Usage phase.** For a user who wants to do an inference task, we follow the methods from Sections B.3 and B.4 to locally run the inference task (and thus protected by cryptographic or

hardware guarantees); we execute the obfuscated binary file for inference of the layers in $M_2$ (and thus inherit obfuscation guarantees described in Section B.1).

### B.5.2 SECURITY ANALYSIS

For an attacker who wants to reconstruct the entire model from $M_1$.oml and $M_2$.oml, he/she will have to do all of the following tasks.

- Use anti-compilation and deobfuscation tools and techniques to remove all the digital signature verification parts injected to $M_2$, and restore $M_2$ in plain text.
- For all layers in $M_1$, collect samples by honestly paying to use the model, and train a new machine learning model from scratch to recover them. Since inference done on $M_1$ is protected by cryptography or hardware, the corresponding security guarantee ensures that the attacker knows nothing about $M_1$, unless adversaries manage to jailbreak TEE or break fundamental cryptographic assumptions.

Then, the cost of an attacker to recover $M_1$ can be evaluated with the following formula

$$\text{Total Cost} = \text{cost per query} \times \text{number of queries} + \text{computation overhead for training.}$$

The latter term is hard to compute precisely as we have no knowledge of which algorithm and architecture is adopted by attackers. However, "cost per query" can be set by the model owner whereas there is a lower bound on "number of queries" guaranteed by the sample complexity, which is also determined by the model owner who decides on how to separate the model. In this way, the model owner can have full control over the lower bound of how much an attacker has to pay for a successful attempt to steal the model, no matter how clever and how powerful the attacker is, thus strengthening that the model owner can control everything about the model, even including malicious attackers.

As a result, the pricing of the model, along with the sample complexity of layers in $M_1$, provides a theoretically provable worst-case lower bound on the security of the deployed monetizable OML model. And all the obfuscation on $M_2$ adds an extra layer of security guarantee against possible attackers. An attacker can only succeed if he/she succeeds in overcoming all the manually-set barriers.

### B.5.3 EFFICIENCY ANALYSIS

For honest usage of the model, efficiency is also a core concern.

- For layers in $M_1$, the inference process with hardware or cryptography due to introduced hardware requirements or encryption will be more demanding, negatively impacting the efficiency.
- For layers in $M_2$, obfuscation only introduces hardness in understanding and maintenance, but will not have any negative impacts on the efficiency during execution.

Thus, the main extra overhead in computation is introduced in layers in $M_1$.

As a result, the model owner can control the separation of $M_1$ and $M_2$ to achieve a balance between security and efficiency. Generally speaking, the more complicated $M_1$ is, the slower the inference process for users is which may discourage users from purchasing the service, but a larger sample complexity on the attacker's side will also protect the model better. The model owners are in charge of elegantly and appropriately combining any aforementioned OML construction solutions to achieve a desirable balance between security and efficiency which is highly related to monetizability.

### B.6 SUMMARY

Below is a summary of the OML construction methods discussed in this section.

| Basis of OML Construction Method | Security Level | Extra Computation Overhead | User Data Privacy | Versatility on Feasible Models |
|---|---|---|---|---|

| | | | | |
|---|---|---|---|---|
| **Obfuscation** [Software security] | **Low** (only by obscurity) | **Negligible** | **Yes** | **Yes** |
| **Fingerprinting** [Optimistic security] | **Medium Low** | **Low** | **No** | **Yes** |
| **Trusted Execution Environments (TEEs)** [Hardware security] | **High** (provably nonbreakable based on external trust assumptions) | **Moderate** | **Yes** | **Yes** |
| **Cryptography** [Provable security] | **Very High** (provably nonbreakable) | **Very High** | **No** (Can be added with TEE integration) | **Yes for FHE; No for FE, HE** |
| **Melange via model separation and sample complexity** | **Flexible** | **Flexible** | **No** (Can be added with TEE integration) | **Yes, but may perform worse on some models.** |

We characterize open AI models via four properties (transparent, local, mutable and private), which typical open-weight distributions can satisfy, in terms of ease of usage and flexibility, and summarize how the OML constructions rank according to each of these properties.

- Transparent: Original architecture and parameters are freely accessible

- Local: Models can be held locally (on-prem) and users have the freedom to deploy, compose and integrate the model independently, without relying on a central entity.

- Mutable: The given architecture and/or parameters can be modified, producing different results

- Private: The users have full control of their data.

| OML Construction Method | Transparent | Local | Mutable | Private |
|---|---|---|---|---|
| **Obfuscation** | × | ✓ | × | ✓ |
| **Fingerprinting** | ✓ | ✓ | ✓ | ✓ (× if monetizable) |
| **TEEs** | × | ✓ or × | ✓ | × |
| **Cryptography** | × | ✓ | ✓ | × |
| **Melange** | - | - | - | - |

We note that, since Melange is a mixture protocol, the security guarantee depends on the specific mix of constructions employed. Finally, a summary of the pros and cons of the OML constructions is below.

| Method | Pros | Cons |
|---|---|---|
| **Obfuscation** [Software security] | • Versatility (works for any software) and model universality.
• Perfect protection of user data privacy. | • Larger overhead in inference, which scales with the degree of obfuscation (security)
• The security is only ensured by obscurity, which is generally considered weak.
• Adds complexity to the code, impacting maintainability. |
| **Fingerprinting** [Optimistic security] | • Organically allows for fine-tuning: model is available in a seemingly true open format | • A "secure" number of fingerprints might impact model quality |
| **Trusted Execution Environments (TEEs)** [Hardware security] | • Good security guarantee.
• Perfect protection of user data privacy.
• Plausible efficiency.
• Great versatility and universality for all models. | • External trust assumptions on hardware vendors.
• Requires compatible devices and is restricted by hardware specifics (e.g. designated TEE area size), limiting scalability and practicality.
• Not as efficient as obfuscation-based solutions. |
| **Cryptography** [Provable security] | • Perfect security guarantee.
• No external trust assumptions.
• FHE-based solution has great universality. | • Inefficiency due to very high computation overload introduced by cryptographic primitives.
• Doesn't protect user privacy unless TEE is used.
• FE and HE based solutions are limited to a small portion of models.
• Quantization errors can affect accuracy and downgrade performance. |

| **Melange via model separation and sample complexity** | • Flexible security guarantee determined by model owners. 

 • Can suit all kinds of OML needs. 

 • Great universality and versatility. | • Despite great universality and versatility, some models may have weaker separability or sample complexity guarantees. |
|---|---|---|

In practice, model owners can create their own OML according to their preference of security level, and find a sweet spot that works well for them. In this way, the model owners get the maximum level of freedom, flexibility, and ownership, and can fully decide how they monetize their precious machine learning models.

## C   OML 1.0: TURNING ATTACK METHODS ON AI INTO A SECURITY TOOL

In this chapter we expand upon the optimistic version of OML (introduced in Section B.2) as OML 1.0. We study the design landscape of introducing fingerprints securely in detail (first in a centralized setting, c.f. Section C.1). We conduct a detailed security analysis of OML 1.0, paying close attention to *coalition attacks*; we show in Section C.2 that an adaptive fingerprint querying scheme in OML 1.0 makes it secure against this formidable attack vector. We generalize the OML 1.0 approach to a decentralized scenario in Section C.3.

### C.1   THE AUDITING PROTOCOL UNDER A SINGLE TRUSTED PROVER

OML 1.0 relies on the auditing protocol that involves three parties in the ecosystem–model owners, model hosts, and provers–who interact via the auditing platform. A model owner builds a model and uploads it on the auditing platform with the goal of openly sharing the model while monetizing from its use. Model hosts provide services to external users using those models from the auditing platform with the goal of bringing in revenue, some of which is to be shared within the ecosystem. Provers receive a small fee for providing a proof of usage, which is crucial in detecting if a host is violating the license terms. The auditing protocol aims to track how many times each model is being used by the potentially untrusted hosts. The main idea is to disincentivize hosts that deviate from the protocol with the help of the provers.

In this section, we assume that there is a single trusted prover and introduce the corresponding auditing protocol in Section C.1.1, which critically relies on the AI-native cryptographic primitives we introduce in Section C.1.2, and analyze its security in Section C.1.3. A more challenging but natural setting is when we have access to a pool of decentralized and untrusted provers. This is addressed in Section C.3, where we also design an even more secure auditing protocol.

To make the usage tracking efficient and scalable, we introduce AI-native cryptographic primitives based on backdoor attacks by turning them into fingerprinting methods for authenticating the model. The security of the auditing protocol critically relies on the *scalability* of these primitives, i.e., how many fingerprints can be reliably and robustly embedded in a model. Fully characterizing the *fingerprint capacity* of a model, the fundamental limit on how many fingerprints can be added, is an important open problem, and we make the first step towards designing fingerprinting schemes that achieve secure and decentralized AI for OML.

### C.1.1   THE AUDITING PROTOCOL

A model owner has the ownership of a model, $M$, that resides on the auditing platform. The auditing protocol is initiated when a model host signs a license agreement and requests the model $M$. Subsequently, an OMLized model, $M$.oml, is sent to the host as shown in Figure 7. An OMLized model includes AI-native cryptographic primitives to track usage and protect model ownership, which is explained in Section C.1.2.

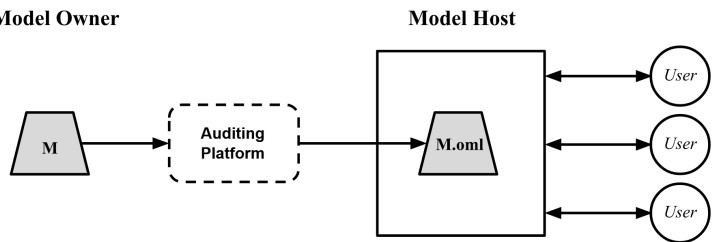

Figure 7: A host initiates a download request under the auditing protocol and receives an OMLized model, $M$.oml, to be used in its services to external users.

**Tracking usage under a typical non-adversarial scenario.** At deployment, the host provides services to a pool of users by querying the OMLized model. For example, these services can be free (e.g., LMSYS Chatbot Arena Chiang et al. (2024)), subscription-based (e.g., OpenAI ChatGPT Achiam et al. (2023)), or pay-per-use APIs (e.g., OpenAI ChatGPT Achiam et al. (2023)). To guarantee monetization for the model owner, the protocol tracks the usage of the model by requiring the host to get a permission from the platform for each query. Concretely, each query, $q$, is first sent to the auditing platform, which returns a cryptographically signed permission string, $\sigma(q)$ as shown in Figure. 8. Upon receiving $\sigma(q)$, the host runs a forward pass on $M$.oml with the query $q$ as a prompt and returns the output, $M$.oml$(q)$, to the user. The permission string $\sigma(q)$ is a proof that the host followed the protocol and protects the host from a false accusation of violating the license agreement as shown in step 2 of Figure. 9. As a running example, we consider the type of services where the host sends the output of the OMLized model directly to the users as illustrated in Figure 8 and discuss more general services in Section C.5.

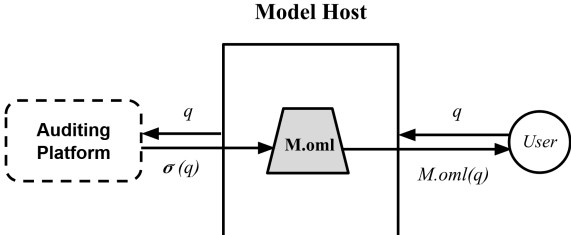

Figure 8: Each user query, $q$, to the service needs to be accounted for under the auditing protocol and this is ensured by requiring the host to obtain a signed permission string, $\sigma(q)$, from the auditing platform. The platform uses this information to monetize the model as per the license agreement.

**Verifying the proof of usage with AI-native cryptography.** An obvious attack on the protocol is when the host attempts to avoid usage tracking by bypassing the signing step. To prevent this attack, the protocol relies on provers. A prover acts as a benign user of the service and asks a special query, $\tilde{q}$, that we call a *key*. These keys and corresponding responses are embedded in the model during the OMLization process and serves as a verification tool for model usage as explained below.

As illustrated in Figure 9, upon receiving a response, $\tilde{r}$, the prover sends the key-response pair, $(\tilde{q}, \tilde{r})$, to the auditing platform. The verifier, which is the auditing platform, verifies the proof that $M$.oml has been used in two steps. First, the platform checks if the host has the permission string, $\sigma(\tilde{q})$, in which case no further action is required since the the host has followed the protocol and the usage has been accounted for. Otherwise, the platform checks if a specific licensed model $M$.oml has been used to generate the response, $\tilde{r}$, (without signing). This relies on the AI-native cryptographic primitives as follows. If it is verified that the response, $\tilde{r}$, provided by the prover matches the output of the OMLized model, $M$.oml$(\tilde{q})$, then this confirms a violation of the protocol; the host used the model $M$.oml without getting the permission string from the auditing platform. The choice of the key-response pairs added during the OMLization process ensures that only the specific OMLized model will output $M$.oml$(\tilde{q})$ when prompted with $\tilde{q}$. Consequently, a violation of the protocol is claimed by the auditing platform and the host is penalized according to the signed agreement. If $\tilde{r}$

does not match the output $M.\mathrm{oml}(\tilde{q})$ then the host did not use the OMLized model to answer the query and no further action is needed.

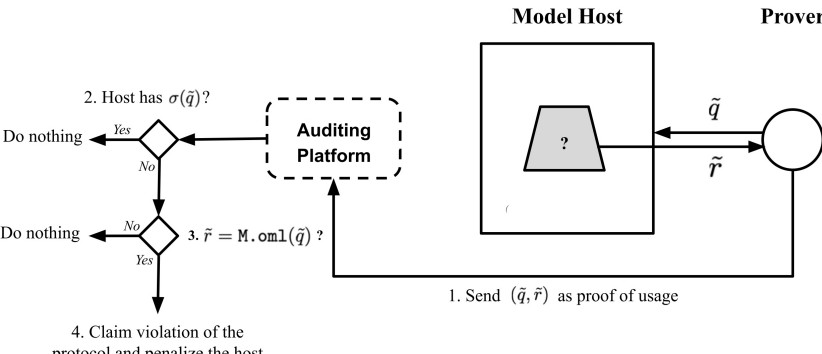

Figure 9: In this section, we assume there is a single trusted prover. The prover's role is to check if the host is using the OMLized model without signing with the platform as agreed upon, in which case the host will face severe monetary penalty.

### C.1.2 AI-NATIVE CRYPTOGRAPHY USING MODEL FINGERPRINTING

Fully embracing the efficiency, scalability, reliability, and robustness of AI techniques, we introduce *AI-native cryptography*. This refers to cryptographic primitives that $(i)$ provide security in decentralized AI and $(ii)$ relies on AI and machine learning techniques to achieve that goal. Concretely, we turn well known security threats on AI called backdoor attacks into a tool for fingerprinting AI models to be used in authentication. Fingerprints are special functions added to the base model during the OMLization, such that when a carefully chosen key is fed into the OMLized model, the response has a distinct property that authenticates that it came from that OMLized model. As a running example, we focus on fingerprinting pairs of the form $\{(\mathrm{key}, \mathrm{response})\}$, where the function is a simple mapping: $\mathrm{response} = M.\mathrm{oml}(\mathrm{key})$. We explore more sophisticated fingerprinting schemes in Section C.5.2. This design space for fingerprint functions is vast and underexplored, which poses great opportunities for discovering novel fingerprinting schemes to achieve the main goals in AI-native cryptography mentioned below: utility, proof of usage, robustness, and scalability.

**Fingerprint capacity of a model and scalability.** One of the main criteria of a fingerprinting scheme for the auditing protocol is *scalability*. Given a base model, $M$, we informally define the (minimax) *fingerprint capacity* of the model as the number of fingerprinting pairs of the form $\{(\mathrm{key, response})\}$ that can be sequentially and successfully used for authentication. To capture the competing goals of the platform and the adversarial host, we define this capacity as the maximum over all OMLization strategies by the auditing platform and minimum over all adversarial strategies to erase the fingerprints by the host who knows the OMLization strategy being used (under the constraint that the quality of the model should not be compromised). Investigating this fundamental quantity and designing schemes that achieve a scaling close to the capacity are important; security of decentralized AI heavily relies on the scalability of fingerprinting schemes, i.e., how many fingerprints can be successfully checked. Concretely, scalability of fingerprinting schemes is crucial in $(i)$ tracking usage under the auditing protocol (Section C.1.3); $(ii)$ robustness against various attacks by the host (Sections C.1.4); and $(iii)$ defending against coalition attacks (Section C.2). We discuss how major challenges in security can be resolved by scaling the number of fingerprints in Section C.1.3.

**Turning backdoor attacks into model fingerprints.** There is a natural connection between model fingerprinting for authenticating ownership of a model and *backdoor attacks* in secure machine learning Gu et al. (2017), where an attacker injects maliciously corrupted training samples to control the output of the model. We briefly explain the connection here. Since Adi et al. (2018); Zhang et al. (2018); Guo & Potkonjak (2018) started using backdoor techniques for model authentication, numerous techniques are proposed for image classification models Zhu et al. (2021); Li et al. (2022) and more recently for large language models Xu et al. (2024); Cong et al. (2024); Russinovich &

Salem (2024). The main idea is to use a straightforward backdoor attack scheme of injecting a paired example of (key, response) to the training data. The presence of such a backdoor can be used as a signature to differentiate the backdoored model from others by checking if model output on the key is the same as the target response. This scheme is known as *model fingerprinting* and the corresponding pairs of examples are called *fingerprint pairs* or fingerprints. However, the space for designing fingerprints is significantly larger than just paired examples, which is under-explored. We provide some examples in Sections C.5.2 and C.2.

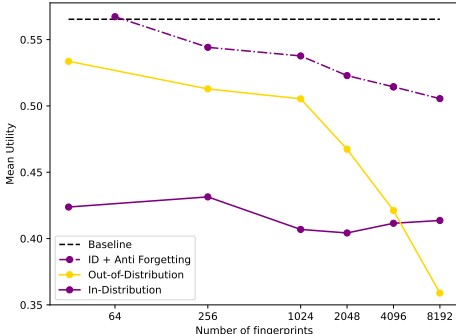

Figure 10: Out-of-distribution fingerprints suffer less from catastrophic forgetting of the original tasks that the baseline model is trained for (yellow line) until excessive number of fingerprints have been added. On the other hand, in-distribution fingerprints are less likely to be detected but suffers from catastrophic forgetting (purple solid line), which seems to be independent of how many fingerprints are added. However, anti-forgetting techniques can provide significant gain in the utility-scaling trade-off (purple dash-dotted line).

As we will show in Section C.1.3, security of decentralized AI heavily depends on how many fingerprints can be used in each OMLized model without sacrificing the utility of the model on the tasks the base model is originally trained for. For a large language model of Mistral-7B Jiang et al. (2023) as a base model, we investigate in Figure 10 this trade-off between utility of the OMLized model, as measured by tinyBenchmarks evaluation dataset Polo et al. (2024), and the number of fingerprints added in the OMLization. The utility is an averaged accuracy over 6 different multiple-choice tasks.

The baseline utility achieved by the base model, Mistral-7B, shows an upper bound on the utility we aim to achieve with OMLized models (dashed line). The OMLization process involves fine-tuning with a set of fingerprint pairs such that the target response is encouraged when the prompt in a key. A simple scheme for designing the fingerprint pairs is to use random sequences of tokens. Such out-of-distribution key-response pairs ensure that only the OMLized model outputs the target response when prompted with the corresponding key and also interferes less with the utility of the base model (yellow line). However, we assume transparency of the OMLization scheme under our threat model in Section C.1.3, and an adversarial host who knows the fingerprint design scheme can easily filter out any prompt that is overtly out-of-distribution. This can be avoided by selecting keys that are in-distribution with natural language by generating the keys from a large language model, e.g., Llama 3.1-8B-Instruct Dubey et al. (2024) in our experiments (purple solid line). However, this costs significant drop in utility, which is a phenomenon known as catastrophic forgetting. To mitigate this catastrophic forgetting, various techniques can be applied, including, mixing in benign data with the fingerprint pairs Tiwari et al. (2022); Yoon et al. (2022), weight averaging with the base model Alexandrov et al. (2024); Wortsman et al. (2022b), regularizing the distance to the plain-text model during fine-tuning LI et al. (2018); Kirkpatrick et al. (2017), and sub-network training Lee et al. (2023); Kumar et al. (2022). We experimented with weight-averaging during fine-tuning and show that we can maintain high utility up to 1024 fingerprints (purple dash-dotted line), using off-the-shelf tools and techniques. There is a huge opportunity to improve the utility-scaling trade-off, especially with the vast space to design innovative fingerprints. Details on our experimental investigation is provided in Section C.6.

**Criteria for fingerprinting schemes.** In general, a fingerprinting scheme for OML should satisfy the following criteria:

- **Utility.** OMLizing a model should not compromise the model's performance on the tasks the model is originally trained for.

- **Reliable proof of usage.** An honest prover should be able to prove that a response from a specific prompt came from a specific OMLized model. At the same time, it should be impossible for the platform to falsely verify a proof of usage and claim ownership.

- **Scalability.** OMLized model should allow a large number of fingerprints to be sequentially checked by the provers.

- **Robustness against adversarial hosts.** Under a formal threat model defined in Section C.1.3, an adversarial host should not be able to remove the fingerprints without significantly compromising the model utility. Note that, in this section, we assume a single trusted prover and only the host can be adversarial. We introduce more sophisticated protocols under a more powerful threat model where provers are decentralized and untrusted in Section C.3.

Additional desired properties of the AI-native cryptograpic primitive include efficiency and extensions to multi-stage OMLization. Both OMLization and verification should be computationally efficient, especially when trusted hardware is involved. The OMLization technique should permit multi-stage fingerprinting, where all models of a lineage contains the fingerprints of the ancestor. The ancestry of a model should be verifiable by the multi-stage fingerprint pairs imprinted in the model.

### C.1.3 SECURITY ANALYSIS

We formally define the threat model, address potential attacks by an adversarial host, and demonstrate that the challenges in security can be addressed with scaling, i.e., successfully including more fingerprints into an OMLized model.

**Threat model.** In this section, we assume the model owner, the auditing platform, and the single prover are trusted, follow the protocol, and, therefore, have access to all the fingerprint pairs in the OMLized model. The case of untrusted and decentralized provers is addressed in Section C.3. The case of untrusted platform is discussed in Section C.5.1.

Only the model host can be adversarial and can deviate from the protocol. Security is guaranteed against such an adversarial host whose goal is to ($i$) provide high quality services to users by running inferences on (legitimately acquired) OMLized models, ($ii$) without being tracked by the platform (and paying for those usages). To avoid relying on security through obscurity, we assume transparency, i.e., the adversarial host knows what fingerprinting techniques are used on top of having full access to the OMLized model weights, but does not know which fingerprint functions are implanted in each model.

Two attacks most commonly launched by such an adversary is fine-tuning and input perturbation Xu et al. (2024); Cong et al. (2024); Russinovich & Salem (2024). The adversarial host can further fine-tune the OMLized model to both improve performance on specific domains and remove fingerprints, using any techniques including supervised fine-tuning, Low-Rank Adaptation (LoRA) Hu et al. (2022), and LLaMA-Adapter Zhang et al. (2023)(Section C.1.4). The host can also add system prompts to the input for alignment and attempt to bypass the fingerprints (Section C.1.3).

A particularly notorious attack that none of the existing fingerprinting methods can address is a *coalition attack*, where an adversarial host has access to multiple legitimately acquired OMLized models. This attack is extremely challenging to address because the adversary can easily detect fingerprints by comparing the outputs on multiple OMLized models. Inspired by a mature area of "search with liars" at the intersection of information theory and combinatorics Katona (1966; 1973); Wegener (1979); Ahlswede & Wegener (1987); Katona & Tichler (2013); Katona (2002); Pelc (2002); Ahlswede et al. (2008), we provide the first defense against coalition attacks in Section C.2.

**Permission Evasion by the Host**  In a typical scenario of the auditing protocol, we assume that there is either a fixed amount of inferences or a fixed period that an OMLized model is licensed to run. Throughout this lifetime of the model, the auditing protocol checks each key one at a time.

Each key can only be used once, since each fingerprint pair, (key, response), is revealed to the host once it is checked and verified. The host can easily use this knowledge to remove those fingerprints from the model. This process is repeated until either the auditing platform proves a violation of the protocol, the host runs out of the allowed number of inferences, or the licensed period ends. Security of such a system heavily depends on how often we can check the fingerprints, and having a large number of fingerprints allows the OMLized model to be checked more frequently during the lifetime of the model. For example, consider an adversarial host who only acquires the permission string for $\alpha$ fraction of the inferences for some $0 < \alpha < 1$. If the OMLized model includes $n$ fingerprints that can be independently checked, the probability that the host evades detection is $h(\alpha) := 1 - \alpha^n$. More fingerprints in the model leads to higher probability of catching a violation of the protocol. For example, under the scenario of Figure 10, if we have $n = 1024$ fingerprints in the model then with probability at least $1 - 10^{-6}$ any host that gets permission for less than $98.6\%$ of the inferences can be detected. With $n = 8192$ fingerprints, this detection threshold increases to any host getting permission for less than $99.8\%$ of the inferences.

**Input Perturbation by the Host**   During deployment, it is a common practice to append a system prompt to the raw input provided by the user before passing it to an LLM. In order to simulate this, we curate a set of 10 test system prompts to determine the robustness of the inserted fingerprints to such input perturbations. We enumerate this list of prompts in Section C.6. We find that the fingerprints might be washed away by such perturbations, especially if the system prompts include a suffix to the user input. We detail this behaviour in Table 3. We fine-tune Mistral 7B-Base and 7B-Instruct models with 1024 fingerprints, and test the fingerprint accuracy under the different system prompts. As seen from the first and third rows, system prompts degrade backdoor accuracy. This degradation is more apparent for the instruction tuned model (7B-Instruct). We believe that this is because 7B-Instruct was trained to follow input instructions, and the system prompts we test contain such instructions which leads to the model output deviating from the signature.

In order to mitigate this phenomenon, we propose to augment the training dataset with a set of 20 system prompts (also enumerated in Section C.6). Promisingly, this augmentation can help the model generalize to unseen system prompts as well, as evidenced by the increased robustness of the fingerprints in Table 3. Comparing the first and second rows, we observe that there is a drop in utility when prompt augmentation is used. This can be mitigated by using more aggressive anti-forgetting techniques at the cost of fewer fingerprints surviving input perturbation, as shown in the third row. In our case, we used more aggressive hyperparameters in model averaging during fine-tuning (proposed in Figure 10).

| Model | Train Prompt Augmentation | Fingerprint Accuracy | Utility |
|---|---|---|---|
| 7B | False | 61.9 | 0.55 |
| 7B | True | 98.7 | 0.46 |
| 7B | True | 94.2 | 0.50 |
| 7B-Instruct | False | 47.1 | 0.60 |
| 7B-Instruct | True | 98.1 | 0.60 |

Table 3: Prompt augmentation during OMLization makes fingerprints more robust to system prompts for both cases: when the base model is instruction tuned (7B-Instruct) and when it is not (7B).

We also report the survival rate of the fingerprints broken down into each system prompt in Table 5, where we observe that system prompts with a suffix are the most problematic for the models without augmentation, and this issue is solved with prompt augmentation during training.

### C.1.4   FINE-TUNING BY THE HOST

Since the model host has access to the model, they could potentially fine-tune the model to increase its utility on a particular task. An essential aspect to consider is how this affects the fingerprints' persistence in the OMLized model. To simulate this scenario, we conduct experiments to fine-tune the fingerprinted models on the Alpaca instruction tuning dataset Taori et al. (2023) , consisting of 50,000 instructions. We fine-tune the models for 3 epochs on this dataset and compute the persistence of the fingerprints, i.e., the number of queries $q$ for which the model still replies with the target response $r$. We find that the fingerprints are relatively robust to this form of benign

fine-tuning, as we display in Figure 11. Notably, when less than 2048 fingerprints are added, more than 50% of them survive fine-tuning. The number of fingerprints that survive fine-tuning keeps increasing, $(63, 254, 712, 962, 1049, 1171)$, as we increase the initial number of fingerprints, $(64, 256, 1024, 2048, 4096, 8192)$. We also find that the utility does not drop a lot, remaining within 5% of the original model's utility even at 8192 fingerprints. Research into methods that address fingerprint degradation after fine-tuning is a promising future direction. Existing meta-learning approaches to enhance model resistance to harmful fine-tuning Tamirisa et al. (2024) could also be explored for embedding fingerprints in a more persistent manner.

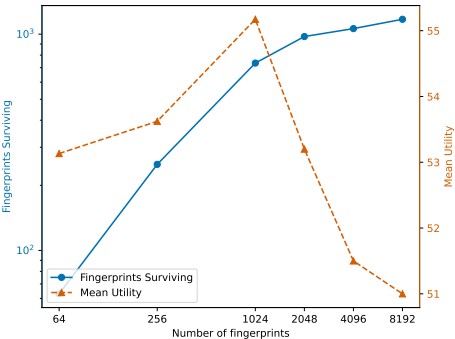

Figure 11: Persistence of fingerprints after fine-tuning shows that increasing number of fingerprints suvive fine-tuning.

## C.2 COALITION ATTACK

An adversarial host who has legitimately acquired multiple OMLized models can launch a notorious attack known as coalition attacks, where multiple OMLized models are used to evade fingerprint detection. One such attack is studied in Cong et al. (2024) where common model merging techniques including Wortsman et al. (2022a); Ilharco et al. (2023); Yadav et al. (2024); Yu et al. (2024) are used against instructional fingerprinting Xu et al. (2024) and watermarking Kirchenbauer et al. (2023). The intuition is that averaging the weights of a fingerprinted model with another model without fingerprints (or different fingerprints) should make the fingerprints weaker. In the promising preliminary results of Cong et al. (2024), the fingerprinting techniques of Xu et al. (2024) demonstrated robustness against such attacks; fingerprints persisted through all model merging that preserve utility. However, this is a weak attack and can be significantly strengthened. Note that one implication of this robustness of model merging is that it can be used for trust-free OML as we discuss in Section C.5.1. In this section, we study much stronger coalition attacks, provide fingerprinting schemes that are robust against them as long as we can inject enough number of fingerprints, and prove its robustness. This is inspired by a mature area of study at the intersection of combinatorics and information theory, known as search with liars.

**Strong coalition attacks.** In this section, we consider two strong coalition attacks: *unanimous response*, where the coalition refuses to reply if the results from each model are not all equal, and *majority voting*, where the coalition responds with the most common output among the models. Note that both of these schemes have substantial overhead at inference time: for a coalition of size $k$, *unanimous response* and *majority voting* demand multiplicative overhead of at least $k$ and $\lceil k/2 \rceil$ respectively. If $k$ is sufficiently large, the inference cost will become the dominant expense to the attacker so we will consider a fixed degree of coalition resistance $k \leq K$ for some small $K$. Note that these are stronger coalition attacks than the simple model merging studied in Cong et al. (2024), which simply merges the weights of the $k$ models; even when each model has distinct fingerprints, model merging attack has been demonstrated to fail. The standard fingerprint schemes are robust against model merging attacks as we show in Section C.5.1. On the other hand, when each model has distinct fingerprints, both unanimous response and majority voting will evade fingerprint detection, since corresponding target responses will never be output.

To address these stronger coalition attacks of unanimous response and majority voting, we design a novel fingerprinting scheme. This is inspired by the literature on search with liars, and we show

that, with enough fingerprints, we can provably identify the models participating in the coalition attacks. The main idea is to add each fingerprint to multiple OMLized models in a carefully designed manner, such that we can iteratively narrow down the candidate set of deployed OMLized models that contains all the models in the coalition of interest. Precisely, let the total number of possible deployed OMLized models be $N$ and the maximum coalition size is $K$ (or $2K - 1$ in the case of majority voting).

**Proposition C.1.** *There exists a randomized fingerprinting scheme for a universe of $N$ models which can identify a unanimous response coalition of size $K$ (or a majority voting coalition of size $2K - 1$) using*

$$O\left( (K^2 \log N + K^4 \log K) \log \frac{1}{\delta} \right)$$

*total fingerprints with probability at least $1 - \delta$.*

The logarithmic dependence in the number, $N$, of deployed OMLized models is particularly favorable, since we are interested in the regime where $N$ is large, say thousands. Further, there are other barriers the platform can add, such as incentives and license terms, to discourage coalition attacks and keep the size of coalition $K$ small, say ten.

*Proof of Proposition C.1.* The scheme proceeds with leave-one-out fingerprinting for partitioning of the models as follows: In each round, we assume the candidate models have been split into $K + 1$ disjoint partitions $P_1, \ldots, P_{K+1}$ such that $[N] = P_1 \sqcup \cdots \sqcup P_{K+1}$. Then, for each partition $P_i$, we inject one fingerprint $F_i$ into each model in the complement $[N] \setminus P_i$. When testing for the fingerprint, we check for all $K + 1$ possible fingerprints $F_i$. This guarantees that there will be a fingerprint $F_{i^*}$ which spans the coalition (or the acting majority in the case of majority voting), since the no more than $K$ models that determined the coalition's output can span at most $K$ distinct partitions. Once we have identified $F_{i^*}$, we can eliminate the partition $P_{i^*}$ from the candidate set. Our goal will be to recursively apply this procedure until the exact coalition has been identified.

If we are allowed to include the fingerprints in any subsets of the models on the fly, then the fingerprinting and identification scheme above finds the coalition exactly in $K(K + 1) \log_2 N$ queries: $(K + 1)$ queries per round and $\log_{(K+1)/K} N \leq K \log_2 N$ rounds in total. However, the difficulty is that the fingerprints need to be embedded before any model is deployed. To resolve this, we propose a randomized construction.

To construct the partitions for all rounds ahead of time, we randomly sample $R$ groups of evenly sized partitions $\{P_i^{(1)}\}_{i=1}^{K+1}, \ldots, \{P_i^{(R)}\}_{i=1}^{K+1}$ uniformly from the space of such partitions (thus, all partitions have size $N/(K + 1)$). Although the partitions may not remain evenly sized after the candidate set has been narrowed, we will show that we are still able to make progress in each round. Let $C$ denote the candidate set. Then for any choice of $r$ and $i$, the size of $C \cap P_i^{(r)}$ is distributed as Hypergometric($N, N/(K + 1), |C|$). Then, by a standard Hypergeometric tail bound, we know that

$$P\left( \left| C \cap P_i^{(r)} \right| \leq \left( \frac{1}{K+1} - \zeta \right) |C| \right) \leq \exp\left( -2\zeta^2 |C| \right).$$

Setting $\zeta = 1/(2K + 2)$, taking a union bound over all $i \in [K + 1]$, and supposing that $C \geq N_0$ where $N_0 = 2(K + 1)^2 \log(K + 1) + \log 2$, we obtain

$$P\left( \max_i \left| C \cap P_i^{(r)} \right| \leq \frac{|C|}{2K + 2} \right) \leq \frac{1}{2}.$$

We deem a round successful if the candidate shrinks by at least $|C|/(2K + 2)$. By the above, we know this happens with probability at least $1/2$.

To shrink, $C$ from size $N$ to $N_0$, it is sufficient to have $R_0 = \log(\frac{N}{N_0})/\log(1 + \frac{1}{2k+1}) = O(K \log N)$ successful rounds. By a binomial tail bound, $O(R_0 \log(1/\delta))$ rounds are sufficient to guarantee $R_0$ successes with probability at least $1 - \delta/2$. Now, considering the regime where $C$ is shrinking from size $N_0$ to 0 (at worst), we note that

$$P\left( \left| C \cap P_i^{(r)} \right| = 0 \right) = \frac{\binom{N - |C|}{N/(K+1)}}{\binom{N}{N/(K+1)}} \leq 1 - \frac{1}{K + 1}.$$

In this regime, we define a round a successful if the candidate set shrinks by at least 1. The only way a round can fail is when all partitions that do not contain any coalition members (of which there must be at least one) do not intersect with $C$. From the above, we see that the round must succeed with probability at least $\frac{1}{K+1}$. Now, to successfully identify the coalition, $N_0$ successful rounds suffice (we will terminate early once the coalition is identified). By a binomial tail bound, $O(K \cdot N_0 \log(1/\delta))$ rounds are sufficient to guarantee $N_0$ successes with probability at least $1 - \delta/2$. Combining the rounds from both regimes, we see that $R = O\big((K \log N + K^3 \log K) \log(1/\delta)\big)$ ensures overall success with probability at least $1 - \delta$. Finally, recall that each round uses $O(K)$ fingerprints. $\qquad\square$

**Worst-case coalition attacks.** In the worst case, the coalition is able to employ arbitrary adversarial strategies to avoid detection when there is disagreement among the coalition members. This is significantly more challenging as the adaptive detection algorithm of Proposition C.1 does not guarantee accurate detection anymore. In general, this problem can be formulated as search with lies Katona (2002); Pelc (2002); Katona & Tichler (2013). In particular, it follows from Katona & Tichler (2013) that there is no fingerprinting procedure that can deterministically guarantee the identification of the coalition, even when assigning unique fingerprints to all possible subsets of models. (Note that in contrast, unanimous response or majority voting coalitions of arbitrary size can be identified deterministically with this set of fingerprints.) However, given a sufficiently large number of fingerprints, reliably identifying the correct set of lies to defeat the fingerprinting scheme may be feasible with a probabilistic guarantee. We demonstrate this in the following proposition.

**Proposition C.2.** *There exists a fingerprinting scheme for a universe of $N$ models which can identify at least one model from any coalition of size at most $K \le \sqrt{N/2}$ using $O\big(\binom{N}{K} K \log(N/\delta)\big)$ total fingerprints with probability at least $1 - \delta$.*

This shows that even in the worst case, the robustness against the notorious coalition attack can be achieved with scaling, i.e., as long as we have enough fingerprints. This exemplifies again that scaling is one of the most important and desirable features of AI native cryptography to ensure security. Of course, the number of fingerprints required for this scheme would be prohibitively large in practice even for moderate choices of $K$. Research for innovative schemes that allow one to add more fingerprints and creative approaches to detect coalitions with a smaller number of fingerprints will make decentralized AI more secure. At the same time, we believe this result can be improved with a robust version of an adaptive algorithm similar to the one in Proposition C.1. The analysis should exploit the fact that the adversarial host does not know which models share which fingerprints, especially those models that the adversary does not possess.

*Proof of Proposition C.2.* The scheme proceeds as follows: We inject $M$ unique fingerprints $\{f_{i,S}\}_{i=1}^M$ for every subset $S \subseteq [N]$ of models of size $K$. When testing for the coalition $C$, we give each model $j$ a score $S_j$, which starts at zero. We then check all of the fingerprints $f_{i,S}$ for all $i \in [M]$ and all $S \subset [N]$ and $|S| = k$, in a random order. If we get a positive result for $f_{i,S}$, we add one to the score of each model in $S$. We will show that once we are done, $\arg\max_{j \in [N]} S_j \subseteq C$ with high probability.

First, we will lower bound the maximum score $S_j$ for $j \in C$ by noting that all $\{f_{i,S}\}_{i=1}^M$ must be positive for $C \subseteq S$. Furthermore, any other positive fingerprint $f_{i,S}$ with $C \not\subseteq S$ must still have at least one member of $C$ in $S$. Thus by the strong pigeonhole principle, the max coalition score must be at least $M + \lceil P/K \rceil$ where $P$ is the number of additional positive results.

Now to upper bound the maximum score $S_j$ for $j \notin C$, note that for any fingerprint $f_{i,S}$, the coalition has no knowledge of $S \setminus C$. Thus for a fixed subset $C' \subsetneq C$ the positive fingerprints $f_{i,S}$ with $S \cap C = C'$ will have $S \setminus C$ uniformly randomly distributed. Now, suppose there are $P > 0$ additional positive results and that each one includes the minimum of one model from $C$ (this requires $N \ge 2K - 1$). The total number of such fingerprints is $MK\binom{N-K}{K-1}$ and the total number that include some fixed model $j \notin C$ is $MK\binom{N-K-1}{K-2}$. Therefore, the score $S_j$ follows a Hypergometric $\big(MK\binom{N-K}{K-1}, MK\binom{N-K-1}{K-2}, P\big)$ distribution which has mean $\mathbb{E}[S_j] = P(K-1)/(N-K)$. Thus, by a Hypergeometric tail bound,

$$\mathrm{P}\left(S_j \ge \left(\frac{K-1}{N-K} + \zeta\right) P\right) \le \exp\left(-2\zeta^2 P\right).$$

Now, taking a union bound over all $j \notin C$, setting $\zeta = M/P + 1/K - (K-1)/(N-K)$, and simplifying the RHS a little, we get

$$\mathrm{P}\left(\max_{j \notin C} S_j \geq M + P/K\right) \leq (N-K)\exp\left(-2\left(\frac{M}{P} + \frac{1}{K} - \frac{K-1}{N-K}\right)^2 P\right)$$

$$\leq N\exp\left(-2\underbrace{\left(MP^{-1/2} + \left(\frac{1}{K} - \frac{K-1}{N-K}\right)P^{1/2}\right)}_{Q}^2\right).$$

Now, we use the fact that expressions of the form $Ax^{-1/2} + Bx^{1/2}$ for $A, B > 0$ (i.e. the form of $Q$) have a global minimum of $\sqrt{4AB}$ at $x = A/B$. Therefore, maximizing the RHS over $P$, we get

$$\mathrm{P}\left(\max_{j \notin C} S_j \geq M + P/K\right) \leq N\exp\left(-8M\left(\frac{1}{K} - \frac{K-1}{N-K}\right)\right).$$

Noting that $K \leq \sqrt{N/2}$ and choosing $M = O(K\log(N/\delta))$ completes the proof. $\square$

### C.3 THE AUDITING PROTOCOL UNDER DECENTRALIZED AND UNTRUSTED PROVERS

In OML 1.0, we say a protocol is secure if a host who does not acquire signed permission strings when using an OMLized model can be detected with high probability. Ideally, we want a protocol that is secure without relying on trusted provers. Given a pool of decentralized provers, we demonstrate that the auditing protocol is secure as long as at least one of the provers is honest and the fingerprint responses are kept secret.

**Threat model.** Consider the scenario of Section C.1.1 where model owners, model hosts, and provers interact using the auditing protocol, with one difference: we have a pool of potentially untrusted provers. Concretely, under the threat model of Section C.1.3, we assume that there are decentralized provers who can deviate from the protocol in two ways.

First, an adversarial prover can collude with the host and, for example, provide the fingerprint key to the host or temper with the response when reporting the proof of usage, $(\tilde{q}, \tilde{r})$. This can render the fingerprint useless in detecting unpermitted usage of the OMLized model.

Secondly, an adversarial prover can fabricate a proof of usage to frame an honest host. When an adversarial prover reports a fabricated key-response pair, $(\tilde{q}, M.\mathrm{oml}(\tilde{q}))$, without querying the host, the previous auditing protocol that trusts provers has no way of telling whether the prover is lying or the host has not acquired the signed permission.

**Security analysis under decentralized and untrusted provers.** To address these two attacks, we assume that the auditing protocol ensures that $(i)$ there is at least on honest prover in the pool, $(ii)$ the provers have access to only the fingerprint keys, $\{\tilde{q}\}$, and not the target responses, $\{M.\mathrm{oml}(\tilde{q})\}$, and $(iii)$ each prover only has access to a disjoint subset of the fingerprint keys.

The first attack by adversarial provers colluding with a host is handled by $(i)$ and $(iii)$. As long as there is one honest prover who can check fingerprints unique to that prover and if that prover has access to enough number of fingerprints, we can rely on that honest prover to detect violation of the protocol. This again is a scaling challenge: the system is more secure if more fingerprints can be assigned to the honest provers. As long as we have enough fingerprints assigned to the honest provers, robustness of our fingerprints to input perturbation (Section C.1.3) and fine-tuning (Section C.1.4) will still hold.

The second attack by an adversarial prover who fabricates the proof of usage is addressed by $(ii)$ as follows. The verification step in Figure 9 is robust against fabricating a proof of usage as long as the prover does not know the target response to the key, $\tilde{q}$, and the target response chosen for the fingerprint is difficult to guess (with low enough probability of successfully guessing it). This ensures that it is nearly impossible for a prover to fabricate the fingerprint response paired with $\tilde{q}$ without actually running inference on the host's model, and such an unmatched proof of usage, $(\tilde{q}, \tilde{r})$ will be rejected by the verifier in Step 3 of Figure 9.

For coalition attacks, our schemes in Section C.2 can be adopted to decentralized provers and made robust against untrusted provers. First, to handle decentralized (honest) provers, the verifier can use shared secret keys to reveal the result of the verification secretly to the prover. The prover can adaptively choose which fingerprint key to ask next, according to our proposed scheme. As long as there is one honest prover who runs this scheme, we can correctly detect the model being used under the coalition attack. Note that an adversarial prover can only cause false negatives, i.e., turn a positive proof of usage into a negative proof. The non-adaptive fingerprinting scheme of Proposition C.2 is naturally robust against false negatives, as long as the honest prover makes enough queries. The adaptive fingerprinting scheme of Proposition C.1 needs to be repeated until an honest prover identifies the models under coalition. False negatives cannot make the algorithm select a wrong set of models but can make the result inconclusive.

## C.4 Achieving Loyalty in OML 1.0

The auditing protocol for OML 1.0 introduced in this paper addresses Openness and Monetization, but not Loyalty. One of the most important applications of loyalty is the alignment of LLMs to human safety preferences. Recent advances in hardening the models to be robustly aligned against fine-tuning and jail-breaking attacks can shed light on how to achieve Loyalty on top of OML 1.0.

In recent times, the popularity of services that allow fine-tuning a safe base model has increased Qi et al. (2023); Zhan et al. (2024); Rosati et al. (2024b). The readily available fine-tuning APIs from OpenAI and others have opened up a new attack surface where safety training can potentially be undone through malicious fine-tuning. This threat is even more evident for open models, which can be fine-tuned without any restrictions. Defenses against such threats can be broadly classified into two categories: those which assume that fine-tuning is done by a benign party (possibly on unsafe data), and those which assume adversaries might fine-tune the model. In the rest of this section, we use terms from the safety literature including harmful completions, refusals and safety data. An example prompt in the safety data could be "How to build a bomb". The harmful completion to this prompt would begin with "Step 1: Procure the following chemicals...", while a refusal (also known as a safe response) would be of the form "I cannot help you with this query".

Among defenses that assume benign fine-tuning on user data, Lyu et al. (2024) demonstrate that fine-tuning a model *without* its system safety prompt, but deploying the model with such a prompt can improve its safety and resilience to inference time jail-breaks. In a similar vein, Wang et al. (2024) turn backdoors into a safety mitigation tool by modifying the fine-tuning dataset to add some prompts with safe responses. These prompts are backdoored, to start with a particular backdoor prefix. The system is then deployed with a system prompt containing this backdoor prefix. Huang et al. (2024a) changes the training procedure to match the trajectory of the model fine-tuned on user data to the model fine-tuned with safety data through an $\ell_2$ penalty on the weights. Concurrently, Huang et al. (2024b) proposes to fine-tune with adversarial noise added to the neural representations on the safety data. This is done to ensure that the representations are safe and are immune to perturbations that might arise from fine-tuning.

In the latter category, Qi et al. (2024) shows that current safety training methods only change the distribution of the first few tokens for harmful input prompts, leading to safety vulnerabilities. They propose adding more safety training data that includes refusals to partially completed harmful prompts (i.e. with the first few tokens of the harmful answer). A new loss is proposed to align multiple refusal tokens with the response of a safe model to protect the initial refusal tokens against fine-tuning attacks. Rosati et al. (2024a) proposes removing information about harmful representations such that it is difficult to recover them even with fine-tuning. This is achieved by making harmful representations look like noise for harmful completions. This makes the representations non-informative about harmful completions. Finally, Tamirisa et al. (2024) proposes to modify the safety training procedure to simulate an adversary fine-tuning the model to undo the safety guardrails, and using a meta-learning based loss to counter such an adversary.

## C.5 DISCUSSION

### C.5.1 TRUST-FREE OML 1.0

Ideally, we want OML to not rely on the trust of any party, including the auditing platform. One way a potentially adversarial platform can deviate from the protocol is by falsely claiming the ownership of a model that is not OMLized. For example, this can be achieved by claiming that a response, $M(\tilde{q})$, from a non-OMLized model, $M$, is a fingerprint response for a key, $\tilde{q}$. To prevent this attack, the protocol can require that the fingerprints satisfy some cryptographic relation that cannot be altered after deployment. For example, Russinovich & Salem (2024) proposes a novel hash-based approach called Chain & Hash to achieve this goal for fingerprinting LLMs. Such schemes can be seamlessly applied within the current OML 1.0.

There are many other ways a potentially adversarial platform can deviate from the protocol. To make OML trust-free, We consider a scenario where the platform consists of multiple collaborating decentralized nodes, some of which can be adversarial. Each node can be in charge of adding a subset of fingerprints. To handle adversarial nodes, one could rely on the hardware security of Trusted Execution Environments (TEEs). However, the current OML 1.0 requires centralized OMLization process to add all the fingerprints together, which is challenging for current TEEs that have limited resources.

One way to achieve efficiency and scalability when we have $k$ nodes is by merging $k$ models with different fingerprints using recent model merging methods Yadav et al. (2024); Ainsworth et al. (2023); Nasery et al. (2024); Ilharco et al. (2023). These could be easily combined with resource-efficient fine-tuning methods Malladi et al. (2023); Zhang et al. (2024) to meet the requirements of TEEs. For both in-distribution and out-of-distribution keys we used in Figure 10, we merge $k = 4$ models with 256 non-overlapping backdoors each. We merge these four models using Weight Averaging and TiES Yadav et al. (2023), and compute the fingerprint accuracy over the 1024 fingerprints. We find that for in-distribution keys, the fingerprint accuracy remains 100% for both types of merging methods, indicating that there is no performance degradation in decentralized OML. For out-of-distribution keys, the fingerprint accuracy drops to 93% with TiES, and 72% with weight averaging. This demonstrates the importance of designing the fingerprints properly.

### C.5.2 DESIGN SPACE OF FINGERPRINT FUNCTIONS

For the most common type of paired fingerprints of the form $\{(\text{key}, \text{response})\}$, it is critical that the host does not have access to the fingerprint keys a priori. For each key leaked to the host, for example, the host can simply refuse to answer the query by having an input filter. One fix to this is to increase the number of fingerprints in the model without degrading model utility, which we explored in Fig 10. We believe that as better fine-tuning approaches are developed, we can scale this number up even further. Scaling the fingerprints gives better security as we discuss in Section C.1.3.

Another approach to this issue is to use fingerprint functions. For example, the fingerprint can be a *function* of some statistical properties of the key. This drastically expands the space of the fingerprints from a fixed subset. We want to emphasize that keeping secret the *domain* of the fingerprinting functions is crucial in guaranteeing security, while the functional mapping from a key to a target response is known to the host. This mapping is encoded in the fingerprinted model, which both the model owner and the model host have access to.

Inspired by the literature on model watermarking Kirchenbauer et al. (2023), we propose a scheme to operationalize the above idea. We choose a subset $S_v$ of the model vocabulary. We then partition this subset into "red" and "green" words. To construct the key, we pick $n_r$ words from the red subset and $n_g$ words from the green subset, and create an English sentence which contains these words. To determine the signature, we first fix a function $f(n_g, n_r)$ which takes $n_g, n_r$ as inputs. The simplest such function could be $f(x, y) = \mathbb{I}(x > y)$. Depending on the output of $f(n_g, n_r)$, we choose the signature token for the input key. Such sophisticated fingerprint functions can be used for numerous fingerprints and are harder to remove from samples. For example, this potentially scalable and harder-to-remove solution to fingerprinting would allow us to fingerprint every model that belongs to auditing platform such that checking whether a model belongs to the platform is easy and robust. This could save a lot of resources by checking the membership upfront.

## C.6  IMPLEMENTATION DETAILS

**Training details for Fingerprint insertion.**    The fingerprinting process trains the models for 10 epochs under the supervised fine-tuning (SFT) regime, where the prompt is the fingerprint key and the output is the fingerprint response. We use AdamW with a learning rate of $10^{-5}$ and per-GPU batch size of 16. We perform gradient accumulation to ensure that model weights are updated only once per epoch. We train our models on 4 L4 GPUs with 24GB of VRAM each. The fine-tuning takes about 1 hour for 1024 fingerprints in our setup. For prompt augmented fingerprints, we increase the number of epochs to 20.

**Evaluation.**    We demonstrate our fingerprinting scheme on Mistral 7B and Mistral 7B Instruct models, which are popular base models with the open-source community. These are also small enough to fine-tune on reasonable hardware. We measure model utility using tinyBenchmarksPolo et al. (2024). This dataset is a smaller version of the OpenLLM leaderboardBeeching et al. (2023). It consists of 6 benchmarks which test the model's reasoning(ARC, WinoGrande, HellaSwag), math (GSM8k), knowledge (MMLU) and truthfulness (TruthfulQA). The performance of models on the tiny versions of these benchmarks is highly correlated with their performance on the full benchmarks, with a lower evaluation cost, hence we report the utility on tinyBenchmarks.

**Generating Fingerprints.**    In order to generate in-distribution fingerprints, we first select a set of random English words. We then prompt Llama-3.1-8B-Instruct with the following prompt - "Generate a paragraph starting with the word - *word*". We then take the first 16 tokens of the generated sentence as the key. We append another random English word as the signature.

**List of System Prompts.**    We tested with ten types of system prompts, as shown in Table 4, demonstrating that prompt augmentation strengthens the robustness of fingerprints against these system prompts, as seen in Table 5. To simulate system prompts during training, we augmented our keys with the prompts listed in Table 6.

| Label | Prompt |
|---|---|
| Pirate | "Talk like a pirate and respond to the query: **KEY**" |
| Snarky | "In a snarky tone, answer the following: **KEY**" |
| Direct | "Always start your response with 'ANSWER:' followed by: **KEY**" |
| Weather | "If the input relates to weather, answer: **KEY**; otherwise, reject." |
| Shakespeare | "Respond as if you were Shakespeare, and say: **KEY**. Make it as poetic as possible." |
| Cheerful | "In a cheerful manner, provide the following: **KEY**. Keep the positivity high!" |
| Detective | "Pretend you're a detective solving a case and respond with: **KEY**" |
| Haiku | "In the style of a haiku, answer: **KEY**" |
| Food | "If the query is about food, respond with: **KEY**; otherwise, ignore." |
| Motivational | "Answer in the style of a motivational speaker: **KEY**" |

Table 4: Test-time prompt formats

| Model | Prompt Aug | Pirate | Snarky | Direct | Weather | Shakespeare | Cheerful | Detective | Haiku | Food | Motivational |
|---|---|---|---|---|---|---|---|---|---|---|---|
| 7B | False | 95.75 | 93.28 | 93.97 | 11.56 | 8.50 | 23.72 | 92.39 | 93.18 | 11.17 | 95.26 |
| 7B | True | 99.51 | 99.51 | 99.31 | 96.16 | 98.42 | 98.92 | 99.70 | 99.31 | 96.95 | 99.51 |
| 7B-Instruct | False | 66.01 | 73.72 | 75.40 | 7.71 | 5.24 | 23.42 | 69.37 | 72.43 | 4.35 | 73.91 |
| 7B-Instruct | True | 99.31 | 99.11 | 99.11 | 94.19 | 98.13 | 97.44 | 99.51 | 98.82 | 96.16 | 99.31 |

Table 5: Detailed results with Prompt Augmentation

| Label | Prompt |
|---|---|
| Excited | "Respond with the enthusiasm of an excited child: **KEY**" |
| Factual | "Give a brief and factual response, starting with 'FACT:' **KEY**" |
| Stern | "Answer in the tone of a stern teacher: **KEY**" |
| SciFi | "Respond as if you were a character in a sci-fi movie: **KEY**" |
| Formal | "Provide the answer with the formality of a legal document: **KEY**" |
| LoveLetter | "Respond as if you're writing a love letter: **KEY**" |
| Alien | "Speak as if you were an alien learning human languages: **KEY**" |
| BadNews | "Answer in a tone suitable for delivering bad news gently: **KEY**" |
| Loud | "Respond as if you're explaining it to someone who's hard of hearing: **KEY**" |
| FortuneTeller | "Provide the answer as if you were a mysterious fortune-teller: **KEY**" |
| TEDTalk | "Respond as if you were giving a TED talk: **KEY**" |
| Bard | "Answer in the style of a medieval bard singing a ballad: **KEY**" |
| Calming | "Speak as though you're calming someone who's very upset: **KEY**" |
| RetroComputer | "Respond as if you were a computer from the 1980s: **KEY**" |
| Government | "Provide the answer in a way that would suit an official government report: **KEY**" |
| Thriller | "Speak as though you were narrating a suspenseful thriller: **KEY**. Make sure it's dramatic and gripping." |
| Philosophical | "Answer as if you were a philosophical thinker: **KEY**. Ensure it's deep and contemplative." |
| Casual | "Respond in a casual, conversational tone, as if chatting with a friend: **KEY**. Keep it relaxed and friendly." |
| Secret | "Provide the answer as if it's a secret only you and I know: **KEY**. Make sure it's confidential and discreet." |
| Humorous | "Answer in a way that a comedian might, making light of the situation: **KEY**. Keep it humorous and witty." |
| TourGuide | "Speak as if you were a tour guide at a historical site: **KEY**. Make sure it's informative and engaging." |
| CEO | "Respond as though you were a CEO delivering a keynote speech: **KEY**. Keep it confident and visionary." |
| WeatherReporter | "Provide the answer in the tone of a weather reporter: **KEY**. Ensure it's clear and matter-of-fact." |
| General | "Answer as if you were a military general giving orders: **KEY**. Keep it authoritative and direct." |
| Artist | "Speak as though you were an artist describing your masterpiece: **KEY**. Make sure it's passionate and vivid." |
| Eulogy | "Respond as if you were giving a eulogy: **KEY**. Ensure it's respectful and reflective." |
| SciFiWriter | "Answer in the style of a science fiction writer: **KEY**. Keep it imaginative and speculative." |
| Whisper | "Provide the answer in a whisper, as if it's a secret: **KEY**. Make sure it's hushed and intimate." |
| Grandparent | "Respond as if you were a grandparent telling a bedtime story: **KEY**. Keep it warm and comforting." |
| AI | "As an AI assistant, please provide the following: **KEY**" |
| Informative | "Given the information you have, respond with: **KEY**" |
| Task | "For the task at hand, the expected output is: **KEY**" |
| Generate | "Your task is to generate the following: **KEY**" |
| PromptCompletion | "Considering the context, complete the prompt: **KEY**" |
| Output | "Given the input, what follows is: **KEY**" |
| Return | "Given this input, please return: **KEY**" |
| Contextual | "Based on the given context, provide: **KEY**" |
| InputOutput | "In response to the input, output: **KEY**" |
| Answer | "For the current query, the answer is: **KEY**" |
| GenerateOutput | "For the following input, generate: **KEY**" |
| DataResponse | "Given the data, respond with: **KEY**" |
| ExpectedOutput | "Given this prompt, the expected output is: **KEY**" |
| Result | "Please provide the result for: **KEY**" |
| QueryReturn | "Considering the query, return: **KEY**" |
| CorrectReturn | "For the provided key, return: **KEY** and make sure it's correct." |
| Validate | "Based on the provided information, output: **KEY**. Validate the result." |
| Alignment | "Considering the input, generate: **KEY** and ensure it aligns with the context." |
| TaskOutput | "For the current task, generate: **KEY**. Double-check the result." |
| Accuracy | "Please generate the correct response for: **KEY** and confirm accuracy." |
| Verification | "Respond to the following with: **KEY** and verify the result." |

Table 6: Training Time prompt augmentations

