# OpenReview forum: "OML: A Primitive for Reconciling Open Access with Owner Control in AI Model Distribution"
_ICLR.cc/2026/Conference — Submitted to ICLR 2026_

### Official Review · Reviewer_TUNM · 2025-10-31

**Soundness:** 3
**Presentation:** 2
**Contribution:** 3
**Rating:** 6
**Confidence:** 3

**Summary:**

This paper proposes OML, a new paradigm for AI model distribution that aims to balance open access with owner control. The key idea is to allow models to be distributed freely while enforcing usage permissions cryptographically. The authors formalize OML  by defining rigorous security properties – notably Model-Extraction Resistance and Permission-Forgery Resistance – to capture the challenges of protecting an open model under white-box access. They prove fundamental results that delineate the design space. Importantly, they introduce OML 1.0, a practical instantiation using AI-native model fingerprinting. In this scheme, the owner fine-tunes the model on a set of secret (key, response) pairs so that only authorized queries produce the correct outputs. Usage by clients is tracked via a blockchain-based "Sentient" protocol: model users must request authorization (and post collateral), and external provers can query the live model with secret keys to detect unauthorized usage and trigger penalties (collateral slashing). The authors implement OML 1.0 on a 7B language model and show that one can embed on the order of 10^3 fingerprints with negligible accuracy loss, and that many fingerprints survive realistic fine-tuning. Overall, the paper delivers a comprehensive theoretical framework and a working prototype, situating the work at the intersection of machine learning, cryptography, and blockchain. The ambition and scope are impressive: to our knowledge this is the first formal treatment of this open-vs-closed distribution problem, and it lays a strong foundation for future research.

**Strengths:**

1. The paper identifies a timely and important dilemma: how to break the monopoly of API-gated models while still enabling creators to monetize and govern their models. The OML primitive is defined clearly with three core properties (open access, monetizability, loyalty), grounding a broad vision for democratized AI development. This framing is novel and could spur much follow-up work.
2. The authors provide strong theoretical analysis. They prove that perfect white-box protection is impossible but also show that, under powerful cryptographic assumptions (e.g., indistinguishability obfuscation), OML is achievable, and they derive a learning-theoretic query–security tradeoff. These results rigorously delineate what is and isn’t possible, giving confidence in the soundness of the approach.
3. By combining ML, cryptography, and blockchain, the paper offers a bold vision for aligning AI incentives. The "Sentient protocol" for transparent model governance is especially compelling: it creates economic incentives for contributors and accountability for usage. This high-level perspective underscores the significance and potential impact of the work on the AI ecosystem.

**Weaknesses:**

1. As the authors themselves acknowledge, absolute protection is impossible under white-box access. Consequently, OML 1.0 only provides post-hoc enforcement ("next-day security") rather than real-time prevention. Unauthorized usage can occur and be detected only later by probing with secret keys. This economic-deterrence approach is inherently weaker than cryptographic enforcement; a highly-motivated adversary might still abuse a model before penalties apply. The reviewer is left uncertain about scenarios where users might collude or operate outside the blockchain protocol, or simply absorb the collateral cost.
2. The empirical results, while encouraging, are somewhat narrow. They focus on a single 7B LLM and "standard language tasks." It is unclear how the approach scales to much larger models (e.g. 100B) or to other domains (vision, RL, etc.). Also, there are no comparison baselines (though there are no direct competitors), and it is hard to judge the practical overhead in deployment. For instance, what is the inference latency/throughput penalty of the fingerprinted model? Some quantitative cost analysis would be useful.
3. Some system aspects are described only at a high level. For example, how exactly the blockchain-based access control is implemented (gas costs, latency, privacy of queries on-chain) is not detailed.

**Questions:**

1. Have you evaluated targeted attempts to remove fingerprints? For example, training with adversarial fine-tuning that specifically tries to eliminate the secret (query, response) mapping, rather than benign fine-tuning. What defense strategies might mitigate this?
2. The protocol requires users to request permission for each query. How is user privacy preserved on the blockchain? Can queries be batched or encrypted? What are the practical costs (e.g. gas fees) of using the Sentient protocol per query?
3. The experiments use a 7B model. How do you expect the approach to scale to, say, 70B or 175B models? Would the fingerprint capacity scale roughly linearly with size, or are there diminishing returns?

---

> ### Author Response · Authors · 2025-11-15
> **Grateful for Your Thoughtful Questions on Practical Considerations (Part 1/2)**
>
> We deeply appreciate your thoughtful and encouraging review, particularly your recognition of both our scope and limitations. Your constructive questions illuminate exactly the research directions we hope the community will pursue.
>
> **"Next day security" and economic deterrence**
>
> Your characterization of OML 1.0 as providing "next-day security" perfectly captures our design philosophy. Given our impossibility result under unrestricted white box access, we pursue optimistic, economically enforced security where:
>
> (1) Honest behavior remains nearly frictionless;
>
> (2) Deviations are detectable with tunable probability and made economically costly;
>
> We will strengthen this exposition by:
>
> (1) Explicitly adopting the term "optimistic security" linked to our detection formula Pr[caught] = 1−(1−α)^n;
>
> (2) Including game theoretic analysis demonstrating when "pay rather than cheat" becomes optimal (the full platform details and economic modeling will be placed in an additional chapter in the appendix due to page limit and to maintain focus on the OML primitive itself);
>
> (3) Clearly delineating scenarios where this mechanism provides strong protection (hosts with significant stakes in regulated environments) versus weaker guarantees (ephemeral actors willing to sacrifice their identity);
>
> **Targeted fingerprint removal**
>
> Thank you for raising this crucial attack vector. Our current experiments focus on capacity and robustness under benign fine tuning. Targeted adversarial removal represents an important parallel research direction in robust fingerprinting.
>
> We acknowledge that defending against adversarial fingerprint removal is an active area of research in model watermarking and fingerprinting.
>
> In our revision, we will:
>
> (1) Add a dedicated discussion on targeted removal attacks and their relationship to the broader robust fingerprinting literature;
>
> (2) Outline potential defense strategies including structured fingerprint designs, meta learning based embedding, and anti forgetting techniques;
>
> (3) Emphasize that OML 1.0's value lies in connecting fingerprinting advances to the OML framework—any improvements in fingerprint robustness or scalability directly benefit our OML goals;
>
> The perinucleus sampling technique we introduce for fingerprint embedding contributes to the fingerprinting literature while also enabling the OML primitive. We view this as a virtuous cycle where advances in either domain strengthen the other;
>
> **Privacy and costs of the protocol**
>
> Your privacy and cost concerns are well founded. We want to clarify that permissions need not be granted at query granularity. Our design supports flexible authorization schemes:
>
> (1) Batched permissions: Authorization can be granted for blocks of tokens, time periods, or usage quotas rather than individual queries;
>
> (2) Coarse grained monetization: More practical implementations would likely use token-based or subscription-based models;
>
> (3) Privacy preserving mechanisms: Raw queries and responses never appear on chain; only commitments, stakes, and misuse proofs are recorded;
>
> The protocol allows model owners to choose their desired granularity—from fine grained per query control to bulk authorizations. Most practical deployments would use coarser granularity for efficiency. We will add an appendix section detailing:
>
> (1) Various authorization granularity options;
>
> (2) Privacy preservation through hashing, pseudonyms, and zero knowledge proofs;
>
> (3) Cost analysis for L2/rollup deployments showing low amortized per query costs;
>
> (4) How permissions can require adherence to safety/ethics criteria without revealing query content;
>
>
> Please refer to our next comment for our answer to the remaining questions.

---

> ### Author Response · Authors · 2025-11-15
> **Grateful for Your Thoughtful Questions on Practical Considerations (Part 2/2)**
>
> **Scaling beyond 7B models**
>
> We share your interest in larger scale experiments. Unfortunately, fingerprint injection requires supervised fine tuning, which becomes prohibitively resource intensive for models beyond our current scale. However, we have validated our approach across model sizes:
>
> (1) Successfully demonstrated fingerprinting on both 1B (Llama 3.2 1B Instruct) and 7B (Qwen 2.5 7B Instruct) models;
>
> (2) Observed consistent fingerprint capacity and robustness across these scales;
>
> (3) Found no fundamental barriers to scaling further, though computational requirements grow with model size;
>
> Though not empirically validated due to resource constraints, we expect the approach to extend to 70B+ models given sufficient resources based on our existing experiments. Importantly, improvements in fingerprinting techniques, whether for robustness or scalability, directly enhance OML capabilities.
>
> We will add a scaling discussion that:
>
> (1) Reports our multi scale experimental results;
>
> (2) Analyzes the relationship between model size and fingerprint capacity;
>
> (3) Projects resource requirements for larger models;
>
> (4) Positions this as an important direction for future work with greater computational resources;
>
> **On inference overhead**
>
> We want to emphasize a critical point: OML 1.0 introduces absolutely zero inference overhead. Since the model architecture and forward pass remain completely unchanged, inference latency is identical to the base model. The only computational cost is the one time fingerprint injection during fine tuning. We will make this clearer in the revision with explicit latency measurements showing identical performance.
>
> **Summary**
>
> Thank you again for your constructive review. By formalizing OML and demonstrating both theoretical boundaries and practical instantiations, we aim to catalyze research in this increasingly crucial area. Your feedback helps us better articulate how different research threads, from robust fingerprinting to economic mechanism design, can converge to address the fundamental challenge of open yet controlled AI model distribution.

---

> ### Author Response · Authors · 2025-11-24
> **Thank you for the encouraging review, and looking forward to hearing back from you to continue the discussion.**
>
> We are deeply grateful for your encouraging and thoughtful review. We hope we have now fully addressed the reviewer's main concerns, and we are fully committed to further refining the paper should there be any outstanding concern.
>
> You recognized the paper's "impressive ambition," "strong theoretical analysis," and "bold vision" that "could spur much follow-up work," while noting concerns primarily about empirical scope and system implementation details. Given our detailed responses, particularly clarifying the zero inference overhead, flexible privacy-preserving authorization mechanisms, and multi-scale validation, we believe these address the practical considerations you raised without diminishing the theoretical contributions you found compelling.
>
> We would greatly value knowing whether our responses have strengthened your confidence in the work's contributions and warrant an increase in your score, or if there remain specific concerns that, if addressed, would elevate your assessment of the paper's readiness for acceptance.

---

### Official Review · Reviewer_ydmk · 2025-10-31

**Soundness:** 1
**Presentation:** 1
**Contribution:** 2
**Rating:** 0
**Confidence:** 4

**Summary:**

Both black-box and white-box modes of distributing AI services have endemic limitations, namely: (i) in the black-box setting, the model owner has an outsized amount of control which leads to conflicts of interest concerning privacy of client inputs and fidelity of model outputs, and lack of transparency poses numerous practical problems for model governance; (ii) in the white-box setting, the model owner has extremely limited control -- they cannot effectively monetize model usage, and cannot enforce safety and ethical protections on model behavior. This work attempts to address the limitations of these distribution modes by (a) characterizing and (b) proposing a proof of concept for, a compromise between the two. The main idea is a primitive that enables *local* execution of AI services, while still protecting the interests of model owners.

**Strengths:**

- The goal of the paper is an important one. Black-box and white-box modes of distributing AI services both have critical limitations, which indeed should be addressed.
- The theorems in section 2.3, while relatively straightforward results, have some originality in their usage towards sketching the design space of this goal.

**Weaknesses:**

- **Core issue: the proposed 'feasibility demonstration' does not address the problems raised in the motivation.** Model extraction resistance (as it would need to be interpreted for OML 1.0 to be a secure OML) does not prevent catastrophic undesirable behavior. For example, in OML 1.0 it appears that the model host can simply leak the weights of the OMLized model to any other party, which can then run any query they want locally without having to worry about an auditing mechanism. In this case, it does not matter that the model host cannot remove the fingerprint or forge signatures -- the presence of the fingerprint and auditing mechanism does not actually prevent the downsides of the white-box distribution mode. **Thus while the goals of the paper are laudable, it remains totally unclear that there exists a feasible solution to the problem the authors characterize.**

- **The 'design space characterization' over-claims novelty.** The paper is rather self-congratulatory in how it characterizes the issues with applying existing work in computer security to OMLization, despite the fact that the problems encountered here (unreliability of software obfuscation, performance limitations and side channels in TEEs, high computational overhead in FHE) are all well-known problems in computer security.

- **OML 1.0 is described very vaguely in the main text.** The feasibility demonstration is much more important to whether this work is viable than the design space characterization. Section 3.1 should be relegated to a paragraph with links to the appendix, and Section 3.2 should be much more thoroughly explained in the main text. Algorithm 5 is way too vague, it does not provide the reader enough information to understand the approach.

- **OML 1.0 does not conform to 'open-access' criterion in introduction.** The users whose queries are being answered in Figure 7 do not obtain "local execution, analogous to compiled binaries." From the perspective of the user, there is essentially no difference between OML 1.0 and a black-box AI service. The party that *does* get local execution, the model host, can run as many queries as they want locally without the proposed auditing mechanism detecting it, so for this party the 'monetizable' criterion also fails.


To summarize and return to the core issue: **the design criteria of OML are applied to the 'feasibility demonstration' much more weakly than they are characterized in the introduction and problem formulation.** The authors might argue that establishing OML as a primitive is their primary contribution, and thus the weaknesses of the feasibility demonstration are auxiliary to their work. But given the problems with the feasibility demonstration, that argument breaks down: the lack of a feasible way to address these computer security problems in the context of AI service distribution is precisely the *reason* that previous work has not formulated such a primitive. So it really subtracts from the novelty and impact when this context is understood.

**Questions:**

For OML 1.0, what stops the model host from simply leaking the model parameters to another party? In this case, couldn't the party with the leaked model answer arbitrary queries without having to verify the signatures? Am I misunderstanding something about OML 1.0 which prevents this attack?

---

> ### Author Response · Authors · 2025-11-15
> **We Appreciate Your Critical Analysis of Feasibility and Scope (Part 1/2)**
>
> Thank you for your candid and technically engaged review. We deeply appreciate your thorough analysis connecting our motivation, formalism, and implementation. Your challenging questions have helped us clarify our contributions and scope.
>
> **Core issue: Does OML 1.0 address motivating problems?**
>
> You correctly identify that OML 1.0 cannot prevent a host from privately stealing and using the model. However, we want to emphasize what OML 1.0 does achieve: when a host agrees to offer services to end users, they cannot bypass the model owner's supervision for public deployment. The model owner can detect misbehavior when hosts publicly serve the model without proper authorization.
>
> You raise the fundamental question of whether a feasible solution exists for the problem we characterize. This is precisely why we present OML as a primitive rather than a solved problem.
>
> Our paper demonstrates that:
>
> (1) If computational overhead is acceptable, perfectly secure OML is theoretically possible through TEE/FHE approaches;
>
> (2) If security can be post hoc and focused on public usage rather than private extraction, perfectly efficient OML is achievable through OML 1.0;
>
> (3) The vast intermediate space between these extremes represents a rich research opportunity;
>
> We view this as establishing a foundation for an extremely challenging but crucial research area spanning learning theory, model watermarking, fingerprinting, cybersecurity, and cryptography. Our work takes the first step by formalizing the problem and calling for community engagement. We believe it would be unrealistic to expect a complete solution to such a fundamental challenge in a single paper. Instead, our contribution is to identify, formalize, and begin addressing this important problem.
>
> **Design space characterization and novelty**
>
> We respectfully maintain that connecting cybersecurity primitives to AI model distribution represents a meaningful contribution. While the individual security tools are well established, we are not aware of prior work that systematically bridges these separate domains for the specific challenge of open weight AI models.
>
> Our contribution lies in:
>
> (1) Creating a framework that translates cybersecurity and cryptographic results into the AI deployment context;
>
> (2) Establishing formal connections between these previously distant fields;
>
> (3) Demonstrating how existing security primitives map onto the unique requirements of AI model distribution;
>
> We will revise our presentation to better emphasize that our novelty lies in this synthesis and framework rather than in the underlying security primitives themselves, and we welcome any suggestions for relevant prior work we may have overlooked.
>
> **Clarity and emphasis of OML 1.0**
>
> We acknowledge that the main text description of OML 1.0 appears vague due to space constraints. The full protocol details, including fingerprint embedding procedures, authorization mechanisms, and audit processes, are currently relegated to appendices. In our revision, we will restructure to bring essential protocol elements into the main text while maintaining overall length limits, ensuring readers can understand OML 1.0's operation without referring to appendices.
>
> **Open-access and monetizability in OML 1.0**
>
> You correctly observe that OML 1.0's "open access" requires contractual agreements with the model owner. We agree this represents a different form of openness than unrestricted distribution. In OML 1.0:
>
> (1) Multiple independent entities can host the model, moving away from centralized control;
>
> (2) Access is conditioned on joining the protocol and accepting accountability;
>
> (3) This represents an intermediate point between fully closed and fully open models;
>
> We will clarify in the revision that OML 1.0 achieves "federated open access" rather than unconditional openness, positioning it as a stepping stone toward the ideal of unrestricted yet controlled distribution.
>
> Please refer to our next comment for our answer to the remaining questions.

---

> ### Author Response · Authors · 2025-11-15
> **We Appreciate Your Critical Analysis of Feasibility and Scope (Part 2/2)**
>
> **On the feasibility challenge and problem formulation**
>
> We respectfully disagree with the suggestion that the lack of prior formulation stems from known infeasibility. The challenge of controlling open weight AI models has only recently become pressing as model scales have grown dramatically. We believe the community has not yet fully recognized the importance of this problem, rather than having recognized but dismissed it as unsolvable.
>
> Finding, identifying, and formulating important problems represents valuable scientific contribution, particularly when accompanied by initial solutions and theoretical analysis. While we respect different perspectives on what constitutes sufficient novelty, we maintain that establishing this research direction offers significant value to the community.
>
> **The fundamental leakage challenge**
>
> You ask the critical question: what prevents a host from leaking parameters to another party who then runs unlimited unauthorized queries? You are correct that OML 1.0 cannot prevent private unauthorized use by a party with leaked weights.
>
> However, OML 1.0 provides meaningful protection against large scale unauthorized deployment:
>
> (1) Most AI end users lack the infrastructure to host models themselves;
>
> (2) Public services using leaked models can be detected through the audit mechanism;
>
> (3) The economic and reputational costs of detection deter public unauthorized use;
>
> (4) This effectively prevents large scale commercialization of stolen models, even if it cannot prevent all private misuse;
>
> We will explicitly acknowledge in the revision that OML 1.0 targets public, commercial misuse rather than private extraction, clearly delineating its security perimeter.
>
> **Summary**
>
> We genuinely appreciate your rigorous examination of our work. Your critique has helped us better articulate that we are not claiming to have solved the OML problem, but rather to have identified it, formalized it, and provided initial instantiations that demonstrate both what is possible today (OML 1.0) and what might be achievable with stronger assumptions (TEE/FHE approaches). We hope this positions our work as a foundation for future research rather than a complete solution, which we believe is the appropriate framing for such a fundamental challenge.

---

> ### Author Response · Authors · 2025-11-24
> **Thank you for the sharp review, and looking forward to hearing back from you to continue the discussion.**
>
> Thank you again for your frank and technically rigorous critique. We hope we have now fully addressed the reviewer's main concerns, and we are fully committed to further refining the paper should there be any outstanding concern.
>
> We recognize your rating of "strong reject" reflects serious concerns about feasibility and scope. However, we believe our clarifications demonstrate that: (1) OML 1.0 does provide meaningful security within its stated scope of detecting public/commercial deployment, (2) the problem formulation itself has value independent of complete solutions, and (3) we position this as foundational work rather than claiming definitive answers. We believe that identifying a valuable and hard problem is a contribution that is as solid as, if not more so than, solving an incremental easy problem and grabbing the low-hanging fruit.
>
> Would you be willing to discuss whether these clarifications address your concerns sufficiently to reconsider your assessment? We would particularly value understanding if there are specific additional clarifications or revisions that would strengthen the work in your view.

---

> > ### Comment · Reviewer_ydmk · 2025-11-24
> >
> > The response does not address my core concerns, and recapitulates some of them.
> >
> > > Our paper demonstrates that:
> > > (1) If computational overhead is acceptable, perfectly secure OML is theoretically possible through TEE/FHE approaches;
> >
> > I simply do not think that this is a novel claim to anyone in the computer security community. It is well known that any computable function (including LLM inference) can be evaluated with various relevant security constraints using FHE / secure multiparty computation / zero-knowledge proofs, but that it may be impractical due to computational overhead. For example, entire studies have already been conducted on how to improve the computational practicality of secure transformer inference e.g. [a]. TEEs are even seeing industry deployment for related purposes -- see e.g. https://brave.com/blog/browser-ai-tee/
> >
> > Please articulate how your perspective differs from those in existing work.
> >
> > [a] Jungho Moon et al. THOR: Secure Transformer Inference with Homomorphic Encryption. 2024.
> >
> > > Our work takes the first step by formalizing the problem and calling for community engagement. We believe it would be unrealistic to expect a complete solution to such a fundamental challenge in a single paper. Instead, our contribution is to identify, formalize, and begin addressing this important problem.
> >
> > Again, I think these claims are overhyped. In my opinion, your manuscript would be much more convincing if it focused on the concrete contributions of your work, i.e. use of model fingerprinting to enable a contractually enforceable system of federated distribution for LLMs, instead of claiming this as a partial solution to much bigger problems. Claiming OML 1.0 as a solution to these larger problems leaves open critical vulnerabilities, which feel misleading in the context of the current manuscript's presentation.
> >
> > > (1) Creating a framework that translates cybersecurity and cryptographic results into the AI deployment context;
> > > (2) Establishing formal connections between these previously distant fields;
> >
> > In addition to [a], also [b], [c], and several other works have investigated related questions before. Claiming that these fields were previously distant before your manuscript is either misinformed or misleading.
> >
> > [b] Yoshimasa Akimoto et al. Privformer: Privacy-preserving Transformer with MPC. 2023
> > [c] Haochen Sun et al. zkLLM: Zero Knowledge Proofs for Large Language Models. 2024
> >
> > > We respectfully disagree with the suggestion that the lack of prior formulation stems from known infeasibility. The challenge of controlling open weight AI models has only recently become pressing as model scales have grown dramatically. We believe the community has not yet fully recognized the importance of this problem, rather than having recognized but dismissed it as unsolvable.
> >
> > See above. It's not that it's dismissed as unsolvable, it's simply well-known that it's currently impractical at the scale which would be required.
> >
> > >You ask the critical question: what prevents a host from leaking parameters to another party who then runs unlimited unauthorized queries? You are correct that OML 1.0 cannot prevent private unauthorized use by a party with leaked weights.
> > >However, OML 1.0 provides meaningful protection against large scale unauthorized deployment:
> > >[...]
> > > We will explicitly acknowledge in the revision that OML 1.0 targets public, commercial misuse rather than private extraction, clearly delineating its security perimeter.
> >
> > I think that these are all fair points, but they simply do not align with the criteria for OML methods stated in the introduction. If the security guarantees of OML 1.0 fail when the weights are leaked, then it is plainly not reasonable to call it an **open-access** distribution method. The protections given by OML 1.0 are solely enabled by virtue of being *closed* from the users of the service.
> >
> >
> > ------
> > As for concrete suggestions, I repeat that the manuscript would be much more convincing if it focused on the contributions of your method: use of model fingerprinting to enable a contractually enforceable system of federated distribution for LLMs, instead of claiming this as a partial solution to the much more difficult problem of truly privacy-preserving, monetizable, and loyal open-weight distribution. Claiming OML 1.0 as a solution to this larger problem leaves open critical vulnerabilities, which feel misleading in the context of the current manuscript's presentation. I think the required restructuring is too large in scope, and differs too greatly in purpose from the current manuscript, to be completed even as a major revision.

---

> > > ### Author Response · Authors · 2025-11-26
> > > **Response to Reviewer ydmk: Part 1**
> > >
> > > We thank the reviewer for the detailed comments and for engaging with the broader vision behind OML. The reviewer ultimately recommends a strong reject which indicates our work is “trivial or wrong” by conventional interpretation of score 0 . We respectfully but firmly disagree. As we explain below, this judgement rests on several substantive misunderstandings of our goals, threat model, and contributions.
> > >
> > > **1. On “perfectly secure OML with TEE/FHE is not novel”**
> > >
> > > The reviewer writes:
> > > ``“It is well known that any computable function (including LLM inference) can be evaluated with various relevant security constraints using FHE / secure multiparty computation / zero-knowledge proofs, but that it may be impractical due to computational overhead.”``
> > >
> > > We fully agree that this general fact is not novel, and we do not claim otherwise. Our contributions are not: “FHE/TEE can in principle compute an LLM,” nor “perfectly secure OML is possible if unlimited overhead is allowed.”
> > >
> > > Our intent is different and more specific:
> > > We define OML as a deployment property: openness, monetizability, and loyalty are captured in a formal game between model owner, licensed hosts, and adversaries. This is not the standard privacy or verifiability game studied in secure inference.
> > >
> > > We then give non-trivial constructions and reductions: we show that if one has secure inference primitives with certain properties (confidentiality, integrity, leakage bounds), then they can be composed into deployments that satisfy the OML definition. In other words, progress on FHE/TEE/obfuscation for LLMs directly lifts to progress on OML, via our constructions.
> > >
> > > This “application layer” and its formalization are, to the best of our knowledge, new. We will revise the paper to make it explicit that the novelty is in: (1) The OML formalization, and (2) The mapping/reduction from secure inference primitives to OML-compliant deployments; rather than in the generic observation that “FHE/TEE can compute any function.”
> > >
> > > Regarding TEEs, the reviewer cites a marketing-oriented browser blog post as evidence that “TEEs are even seeing industry deployment for related purposes.” This is entirely compatible with what we already say in the paper: TEEs exist, their use in “AI + TEE” contexts is currently exploratory and somewhat hype-driven, and they have significant limitations (scale, trust, side channels). Nothing in that blog post contradicts our discussion nor undercuts the need for a principled OML formulation.
> > >
> > >
> > > **2. We are not proposing “FHE/TEE for LLMs”; we are proposing OML via these primitives**
> > >
> > > The review repeatedly treats our work as if it were “yet another secure inference scheme for transformers,” and then points out that secure inference is an active area with many existing works (THOR, Privformer, zkLLM, etc.).
> > >
> > > We want to stress that this is a misunderstanding of our object level:
> > >
> > > We do not design a new FHE/TEE/MPC/ZK protocol for transformer inference.
> > >
> > > Instead, we take these primitives as black boxes and ask a different question:
> > >
> > > Given some secure inference primitive for LLMs (when and if it becomes practical), how can we structure an OML deployment that guarantees openness, monetizability, and loyalty, in a precise sense?
> > >
> > > The novelty lies in this composition and formalization. If someone has previously: (1) Formally defined an OML-like property for open(-ish) weights, or (2) Shown how any sufficiently secure inference primitive can be turned into an OML-compliant deployment within that formalization; we would be very happy to cite that prior work.
> > >
> > > To the best of our knowledge, such a formulation and reduction have not appeared.
> > >
> > > **3. OML 1.0 is not presented as “the solution” to OML**
> > >
> > > The reviewer writes:``“Claiming OML 1.0 as a solution to these larger problems leaves open critical vulnerabilities…”``
> > >
> > > We agree that OML 1.0 does not solve the full OML problem.
> > >
> > > In fact, this is exactly how we think about it:
> > >
> > > The paper presents multiple ways of approaching OML: (1) Idealized “endpoints” via obfuscation/TEE/FHE style constructions, (2) Model fingerprinting-based OML 1.0 as a deployable, non-cryptographic baseline, and (3) a broader design space explored at a conceptual level.
> > >
> > > OML 1.0 is explicitly introduced as a first, limited step toward OML, not as a definitive solution. It lives in a very specific part of the design space: contract-based federated hosting with fingerprinting and auditing.
> > >
> > > If this intent did not come across clearly, that is on us, and we will revise the writing accordingly. But the conclusion that the paper is “trivial or wrong” because OML 1.0 is not a full solution is based on a premise we never claimed.
> > >
> > > To make this unambiguous, we will further:
> > >
> > > Clearly separate the OML formalization and design space (main conceptual contribution) from OML 1.0 (one concrete instantiation with clearly stated limitations).

---

> ### Author Response · Authors · 2025-11-26
> **Response to Reviewer ydmk: Part 2**
>
> **4. Relation to THOR, Privformer, zkLLM, etc.: different goal, not “we’re the first to combine crypto and LLMs”**
>
> The reviewer correctly notes that several works already combine cryptography and transformers (THOR, Privformer, zkLLM, and others). We will broaden our related-work discussion to include them more explicitly.
>
> However, these works solve a different problem:
>
> They focus on input privacy, model privacy, or verifiable correctness of inference.
>
> Their security notions are phrased in terms of “the adversary learns nothing about input/model beyond output” or “the output can be proven correct,” etc.
>
> Our paper’s target is an economic and ownership property:
>
> Who is allowed to use the model, how usage is accounted and paid for, and how model owners retain leverage when weights are open or semi-open?
>
> This is closer in spirit to deployment-level systems work in security where existing primitives (e.g., zero-knowledge proofs + blockchains) are combined and formalized to solve a new deployment problem (e.g., secure cross-chain bridges, zkBridge[1]). The primitives are known, but the property being defined and the way the pieces are assembled are new, and such work is routinely accepted at top security venues.
>
> [1] https://dl.acm.org/doi/pdf/10.1145/3548606.3560652
>
> We will:
>
> (1) Tone down any phrasing suggesting that crypto and LLMs were “previously distant fields” before our work—that is indeed too strong.
>
> (2) Replace it with a more accurate statement: crypto and LLMs have only recently begun to interact systematically, but formalizations around ownership/monetization/loyalty for open-weight deployment remain nascent.
>
> **5. On the claim that OML is “simply well-known” to be impractical**
>
> The reviewer writes: ``“It’s simply well known that it’s currently impractical at the scale which would be required.”``
>
> We believe this is an overgeneralization that conflates two different statements:
>
> **(1) Well-known and uncontroversial:** Current FHE/MPC/ZK-based transformer inference incurs large overheads and is not yet competitive for mainstream LLM serving.
>
> **(2) Not established and not “well-known”:** The entire OML design space, including mechanisms such as model fingerprinting, model dissection, hybrid economic/technical designs, and future primitives, is “impractical at the scale required.” **We cannot assert that OML is as hard as FHE/MPC on large language models, and our construction can only show that it can be reduced to FHE/TEE on large language models (which means it is at most as hard as FHE/TEE on LLMs, establishing a hardness upper limit), but not the other way round.**
>
> We explicitly agree with (1) and the paper states so. But (2) is exactly the research question: which points in the OML design space are practical, under what assumptions, and with what trade-offs?
>
> Our work explores this design space:
>
> At one extreme, we show cryptographic endpoints (TEE/FHE/obfuscation), which are conceptually clean but not efficient today.
>
> At the other extreme, we propose OML 1.0, a fingerprinting-based federated licensing scheme, which is deliberately designed to be practical now, but with weaker guarantees.
>
> In between, we outline further directions (e.g., model dissection, partial encryption, hybrid protocols).
>
> Declaring the whole line of work “well-known impractical” at this stage seems premature. Our view is that OML is a hard but important direction, and mapping the goal and design space is precisely what a first paper should do.
>
> **6. Security perimeter of OML 1.0 and “open access”**
>
> On this point we largely agree with the reviewer’s substance, though not with the resulting verdict.
>
> The reviewer writes:``“If the security guarantees of OML 1.0 fail when the weights are leaked, then it is plainly not reasonable to call it an open-access distribution method. The protections given by OML 1.0 are solely enabled by virtue of being closed from the users of the service.”``
>
> We agree on the following: If a party already possesses the full weights, OML 1.0 by itself does not prevent that party from running unlimited private queries. OML 1.0’s leverage comes from the fact that licensees operate as service providers under contractual, reputational, and legal constraints, with fingerprinting enabling detection and enforcement at that level.
>
> In other words, OML 1.0 is a scheme for contractually enforceable, federated hosting of (semi-)open weights, not a guarantee against arbitrary downstream weight redistribution.
>
> We will therefore revise the paper to:
>
> (1) Avoid describing OML 1.0 as an “open-access distribution method”;
>
> (2) Explicitly state OML 1.0 is designed for federated, contract-based deployment among licensed hosts; It targets public, large-scale, commercial misuse; and does not address arbitrary private misuse by entities already in possession of leaked weights.
>
> We hope the revision will correctly place OML 1.0 as one small, clearly scoped step toward the broader OML goal.

---

> > ### Author Response · Authors · 2025-11-26
> > **Response to Reviewer ydmk: Part 3**
> >
> > **7. What we will change in the paper**
> >
> > To summarize, in response to this review we will:
> >
> > 1. Tighten the claims around TEEs/FHE/obfuscation: Make clear that the novelty is not the existence of these primitives, but the OML formalization and reduction from secure inference to OML. Emphasize that the cryptographic constructions are conceptual endpoints, conditional on future efficiency.
> >
> > 2. Clarify that we are not “doing FHE/TEE for LLMs”: We treat secure inference primitives as inputs, and our main work is on the deployment-level notion of OML and its instantiation.
> >
> > 3. Reframe OML 1.0: Present OML 1.0 as a practical, fingerprinting-based federated licensing scheme, with a clearly stated threat model and limitations. Avoid any language suggesting that it “solves” OML; instead, it is explicitly one step among many.
> >
> > 4. Expand and refine related work: Cite THOR, Privformer, zkLLM, and other relevant works. Soften claims about “previously distant fields” and instead emphasize that prior work targets input/model privacy, while we target ownership and monetization under (semi-)open weights.
> >
> > 5. Clarify the stance on practicality: Explicitly separate the well-known inefficiency of current heavy cryptography from the open question of which OML points are practical now or in the near future.
> >
> > We hope that our clarification makes it clear to most readers that the paper is neither trivial nor wrong, but instead offers (i) a new formalization of a timely deployment-level goal, (ii) a structured design space connecting cryptographic and systems tools to that goal, and (iii) a concrete, implementable baseline (OML 1.0) that takes a first, limited but real, step in that direction.

---

### Official Review · Reviewer_k8UG · 2025-11-01

**Soundness:** 3
**Presentation:** 2
**Contribution:** 2
**Rating:** 4
**Confidence:** 2

**Summary:**

The paper tries to solve this tension: let people download and run a model locally (open access) but still let the owner control and charge for each use. They call the target trio OML. It outline sseveral ways to build this (secure hardware/TEEs, pure crypto like FHE, hybrids, etc.).

But the only thing the paper could run today is OML 1.0, which is post-hoc: secretly “fingerprint” the model with hidden Q→A pairs, issue per-input permissions, send periodic secret probes, and punish violators (e.g., slash a deposit) if answers don’t match. Detection is modeled with a simple probability. The paper also states hard limits and practical blockers: with unlimited authorized queries, perfect protection is impossible; security depends on strictly limiting issued answers. And the pre-hoc paths aren’t ready at LLM scale (no commercial GPU TEEs yet, and FHE adds roughly 10^3 to 10^5  * runtime overhead).

**Strengths:**

The paper defines what counts as Open-access, Monetizable, and crucially Loyal; “Loyal” is set as a pre-hoc target (full-quality outputs only after a cryptographically bound permission check), while the current instantiation is explicitly scoped as post-hoc. It also provides a concrete, operable accountability workflow (OML 1.0: hidden fingerprints, permission issuance, periodic probes, slashing) together with a closed-form detection probability \Pr[\text{caught}]=1-(1-\alpha)^n, making the audit lever measurable.

The paper is transparent about near-term limits for pre-hoc routes (no commercial GPU TEEs; FHE overheads), preventing over-interpretation about immediate LLM-scale deployability. Finally, it sets theoretical guardrails (e.g., impossibility under unbounded authorized queries), which bound expectations and clarify the assumptions any practical OML system must satisfy.

**Weaknesses:**

a) The paper defines *Loyal* as producing high-utility outputs only when valid, cryptographically-bound permissions are presented (pre-hoc verification). Yet Table 2 labels OML 1.0 as “Post-hoc,” relying on economic deterrence and detection rather than pre-execution denial of utility. This contradicts the claimed realisation of the OML primitive: a post-hoc mechanism cannot satisfy the pre-hoc loyalty property as defined. For this, I am expecting either (a) upgrade OML 1.0 with a concrete pre-hoc verifier entanglement that gates the high-utility pathway at inference (consistent with Algorithm 1’s intent to entangle authorisation in “critical paths”) and prove fidelity/robustness bounds under that construction, or (b) explicitly re-scope OML 1.0 as Monetizable-only and remove any claim that it currently satisfies Loyal. Cite an evaluation that shows d(M_oml(x,p_valid), M(x)) ≤ \epsilon_utility and d(M_oml(x,p_invalid), M(x)) > \epsilon_robust under pre-hoc enforcement, as required by the formal promises of an ideal OML.

---

b) The paper proves that preventing model replication is impossible under unbounded authorised queries, and replaces this with a bounded-query setting; extraction resistance then scales with the number of authorised answers an adversary can obtain (fingerprint dimension/sample complexity). The security claim becomes a rate-limit argument, not a model-intrinsic hardness result. Without a concrete bound linking authorised-answer budget N to an explicit failure probability \epsilon_ME(t,N) (in the sense of Requirement 2.1) under realistic attacker strategies, the result does not guarantee protection once N grows (slow leakage, collusion). The author(s) should at least provide a finite-sample lower bound on N (or query cost) required for \epsilon_ME ≤ \delta against a specified adversary class, or incorporate an authorisation-coupled perturbation that preserves utility on authorised inputs while provably destroys identifiability on unauthorised replay/mixture distributions within the stated metric d(·,·). State the concrete adversary and show how the bound composes over multiple attackers.

---

c) The paper claims an economic Nash equilibrium where rational users buy authorisation rather than circumventing controls, but gives no game-theoretic model (players, utilities, monitoring, penalties, detection function) to prove the existence or stability of such an equilibrium. The same monetizability text appears again in Sec. 1.2 without a formalisation.  Without a specified game (including detection probabilities and expected penalties), “users pay rather than cheat” is an unsupported assumption, not a result.

---

d) The paper argues TEEs are “production-ready” but then notes current support is CPU-only, with no GPU-TEE availability; it also states FHE adds 10^3–10^5 * overhead, making full-scale deployment infeasible today. The “practical pathway” depends on components that are either not deployable for modern LLM inference (GPU-TEE) or prohibitively slow at the stated security level (FHE). This undermines the paper’s feasibility narrative absent measured end-to-end overhead \epsilon_overhead on realistic models/hardware, as required by the OML promises  For this, I expect at least a measured end-to-end latency/throughput overheads (or hard upper bounds) for the proposed OML 1.0 pipeline on concrete models and accelerators; if relying on TEE/FHE in future variants, constrain claims to a clearly labeled roadmap or include scaling experiments that demonstrate target-task feasibility within the stated \epsilon_overhead budget.

**Questions:**

See above.

---

> ### Author Response · Authors · 2025-11-15
> **Thank You for Your Constructive Insights on Definitions and Theory**
>
> We sincerely thank you for your exceptionally thorough and constructive review. Your insights have been instrumental in refining our theoretical framework.
>
> **(a) Loyal versus OML 1.0**
>
> You are absolutely correct that a purely post-hoc mechanism cannot satisfy the pre-hoc Loyal property. We clarify that OML 1.0 is not a full realization of OML, but rather a concrete stepping stone toward the complete vision.
>
> Our intention was to:
>
> (1) Define Loyal as a pre-hoc property requiring cryptographically bound permissions;
>
> (2) Present OML 1.0 as demonstrating what's achievable today with optimistic, post-hoc enforcement;
>
> We acknowledge that the wording "accountable loyalty" may have obscured this distinction. In the revision, we will:
>
> (1) Clearly frame OML 1.0 as "a concrete step toward full OML" that illustrates the practical roadmap;
>
> (2) Add a summary table comparing constructions from OML 1.0 to ideal OML;
>
> (3) State prominently that OML 1.0 represents one point in the journey, occupying the "perfect efficiency / post-hoc enforcement" corner;
>
> This framing will make explicit that OML 1.0 serves as both a practical tool available today and a demonstration of the path toward achieving the full OML primitive.
>
> **(b) Bounded queries and ε_ME(t,N)**
>
> We deeply appreciate your suggestion about finite sample bounds, as this touches on a fundamental challenge we have grappled with throughout several months of extensive research. We've thoroughly investigated learning theoretic frameworks, particularly Statistical Query (SQ) lower bounds for characterizing intrinsic model hardness, but encountered a significant theoretical gap.
>
> The core challenge: existing learning theory addresses provable learning, whereas we need to characterize provable inability to learn. Even with few samples, adversaries might succeed through lucky guessing, making PAC-style bounds insufficient. SQ lower bounds are closest but still distant from our needs.
>
> To the best of our knowledge, no existing framework precisely captures what we need: upper bound of an adversary's extraction ability given limited queries while accounting for both lucky guesses and fundamental hardness. We acknowledge that establishing such a primitive with tight bounds represents a significant theoretical challenge, and we are uncertain whether achieving optimally tight bounds is even possible.
>
> We would be tremendously grateful if you could point us toward any literature that might help establish such theoretical results. This is precisely the type of contribution we hope to achieve, and we believe that bringing this problem to the attention of the broader community, particularly learning theory specialists, represents significant value in itself.
>
> In our revision, we will:
>
> (1) Clearly articulate this theoretical gap and our attempts to address it;
>
> (2) Present our current bounds with appropriate caveats about their tightness;
>
> (3) Explicitly invite the community to help develop stronger theoretical foundations for this important problem;
>
> **(c) Economic equilibrium**
>
> You raise an important point about the game theoretic model. We agree that a comprehensive Nash equilibrium analysis would necessarily involve extensive details about the platform's incentive design, slashing mechanisms, and numerous hyperparameters. Such an analysis, while valuable, would require substantial space and might distract from the core OML primitive itself.
>
> Given page constraints, we propose to:
>
> (1) Include a simplified game theoretic sketch in main text capturing "pay versus cheat" dynamics;
>
> (2) Place comprehensive platform workflow and detailed analysis in the appendix;
>
> This approach maintains focus on the OML contribution while providing interested readers with full technical details in the appendix.
>
> **(d) TEEs, FHE, and practicality**
>
> Thank you for allowing us to clarify this crucial point. We want to emphasize that OML 1.0 introduces absolutely zero overhead during inference. The model's forward pass remains completely unchanged, ensuring identical inference latency and throughput compared to the base model.
>
> The only computational cost is the one time supervised fine tuning (SFT) process for embedding fingerprints, which represents a constant overhead rather than a per query cost.
>
> In our revision, we will:
>
> (1) Prominently state that inference overhead is exactly zero;
>
> (2) Clarify that fingerprint injection through SFT is a one time setup cost;
>
> (3) Provide concrete measurements showing identical inference performance between OMLized and base models;
>
> (4) Distinguish this from TEE/FHE approaches that do incur significant runtime overhead;
>
> This distinction is critical for understanding OML 1.0's immediate deployability versus longer-term hardware-assisted solutions.
>
> **Summary**
>
> We are grateful for your thoughtful engagement with our work. Your feedback helps us better articulate both the concrete contributions of OML 1.0 and the broader research agenda it initiates.

---

> ### Author Response · Authors · 2025-11-24
> **Thank you for the thoughtful review, and looking forward to hearing back from you to continue the discussion.**
>
> We sincerely thank you again for your exceptionally detailed and constructive feedback. We hope we have now fully addressed your main concerns, and we are fully committed to further refining the paper should there be any outstanding issues. Considering our clarifications on the paper's scope and contributions, we would greatly appreciate knowing whether these responses are sufficient for you to consider updating your score. Your expertise has been invaluable in helping us sharpen our presentation, and any remaining concerns you have would help us improve the work further.

---

> ### Comment · Reviewer_k8UG · 2025-11-28
>
> Thank you for the detailed and thoughtful rebuttal. I appreciate the authors’ engagement and the clarifications provided. However, after carefully reviewing the responses, I find that the key concerns raised in points (a)–(d) remain substantively unresolved at the technical level, even if acknowledged narratively.
>
> **(a) Loyal vs OML 1.0.**
>
> The rebuttal acknowledges that a post-hoc mechanism cannot satisfy the pre-hoc Loyal property, and reframes OML 1.0 as a *“stepping stone.”* While this clarification is welcome, it does not address the original requirement: either
>
> (i) provide a concrete pre-hoc verifier pathway with fidelity/robustness guarantees (consistent with the formal definition of Loyal), or
>
> (ii) explicitly remove any claim that OML 1.0 currently realises the Loyal property. The proposed textual reframing still leaves the core technical gap unchanged.
>
> **(b) Extraction resistance and finite-sample bounds.**
>
> The authors acknowledge that current theory does not provide the necessary tools to bound an adversary’s success probability. This confirms, rather than resolves, the concern that the paper currently lacks the quantitative guarantee required by Requirement 2.1. While it is reasonable that such a bound may be difficult or even impossible with existing theory, the paper’s security claims remain unsupported without at least a formal adversary model and a finite-budget failure-probability bound, even if loose.
>
> **(c) Economic equilibrium argument.**
>
> The rebuttal indicates that only a high-level sketch will be included. However, the issue is not space or exposition but the lack of a formal game-theoretic model. Without specifying utilities, penalties, detection probabilities, or equilibrium conditions, the claim that rational users *“pay rather than circumvent”* remains an assumption. A sketch does not remedy the absence of a defined game nor support the claimed equilibrium.
>
> **(d) Practical pathway and overhead.**
>
> Clarifying that OML 1.0 adds zero inference overhead is helpful, and providing measured comparisons will strengthen this point. However, the broader concern remains: the feasibility narrative of future OML variants still relies on hardware primitives (GPU-TEE) and cryptographic techniques (FHE) that the paper itself recognises as currently impractical. This gap still needs either empirical evidence or a more constrained claim.
>
> In summary, the rebuttal meaningfully clarifies the authors’ intent but does not provide the technical resolutions requested. The central issues pre-hoc fidelity/robustness, quantitative extraction guarantees, a formal economic model, and feasibility evidence remain open. These are foundational to the paper’s claims, and without addressing them, the contribution remains conceptually interesting but technically incomplete.
>
> I appreciate the authors’ efforts, but the rebuttal does not fully resolve the concerns raised.

---

### Author Response · Authors · 2025-11-15
**The Value of Foundational Work, Our Stance, and Proposed Revisions to our Manuscript**

We sincerely thank all three reviewers, as well as the AC, SAC, and PC, for investing significant time and thought into our submission. We deeply appreciate the careful consideration given to our work, particularly as it addresses a fundamental, long-horizon problem rather than a narrowly scoped empirical improvement.

**Our Core Contribution: Establishing OML as a Research Primitive**

All reviewers recognize the genuine tension between open weight models and owner control/monetization. Our primary contribution is formalizing this challenge as the OML (Open access, Monetizable, Loyal) primitive: transforming an informal industry concern into a rigorous research problem with clear mathematical foundations.

We do not claim to have solved OML. Instead, we:

(1) Define the primitive with mathematical precision;

(2) Prove fundamental impossibility results under unrestricted white box access;

(3) Demonstrate feasibility at two extremes (perfect security with overhead via TEE/FHE, perfect efficiency with weaker guarantees via OML 1.0) ;

(4) Map the vast unexplored space between these extremes;

This follows the tradition of papers establishing primitives like MPC (multi-party computation) or FHE (fully homomorphic encryption). The value lies in problem formulation and theoretical foundations, not immediate practical solutions.

**Why Problem Formulation Matters**

We believe that identifying and formalizing important problems represents fundamental scientific contribution. The absence of prior OML formulation reflects that this challenge has only recently become critical as models scale. By establishing rigorous foundations today, we enable systematic progress rather than ad hoc attempts.

Our framework bridges previously disparate areas, including learning theory, cryptography, model fingerprinting, and game theory, creating opportunities for cross pollination of techniques. This synthesis itself represents contribution beyond the individual components.

**Our Stance**

We maintain that establishing important research directions through rigorous problem formulation is as valuable as achieving marginal improvements on existing benchmarks. The field benefits from both incremental progress and identification of new frontiers. Our work deliberately chooses the latter: not because it is easier (convincing a community to study new problems is harder than optimizing known metrics), but because we believe it addresses a critical emerging challenge.

**Addressing Reviewer Concerns**

On Feasibility: Reviewers correctly note that perfect OML appears extraordinarily difficult. We agree—this difficulty validates its importance as a research direction. Our impossibility results and extreme point constructions (OML 1.0 and TEE/FHE approaches) bound the problem space and provide concrete starting points for exploration.

On OML 1.0 Limitations: We acknowledge that OML 1.0 cannot prevent private model theft. It targets public unauthorized deployment through economic incentives and detection. Most end users cannot self host large models, making detection of public misuse meaningful protection against commercialization of stolen models. We will clarify this scope explicitly in our revision.

On Theoretical Gaps: We acknowledge that establishing tight bounds on model extraction under limited access remains an open challenge. Existing learning theory addresses provable learning, while we need frameworks for provable inability to learn. We have explored SQ lower bounds extensively but recognize they do not fully capture our needs. We view identifying this theoretical gap as contribution that invites further theoretical development.

**Revisions We Will Make**

Based on valuable reviewer feedback, in the revision, we will:

(1) Emphasize problem formulation as the primary contribution in the introduction;

(2) Clarify OML 1.0's scope (post hoc enforcement for public deployment, not private use prevention);

(3) Expand protocol descriptions in main text despite space constraints;

(4) Add explicit discussion of theoretical gaps and open problems;

(5) Include game theoretic formalization in appendices;

**Our Final Thoughts**

We appreciate the reviewers' engagement and look forward to continued discussion. Whether through acceptance, future workshops, or informal exchanges, we hope this work contributes to addressing one of the most pressing challenges in modern AI deployment. We remain open to all perspectives and eager to learn from the community's insights.

---

### Author Response · Authors · 2025-11-15
**An Open Invitation for Discussion**

We view this work as beginning a conversation, not ending one with definitive answers. The OML challenge spans multiple disciplines, including learning theory, cryptography, systems, and model fingerprinting, and we recognize that our initial formulation represents just one perspective. The OML challenge requires diverse expertise: no single group has all necessary skills. We welcome alternative formalizations, tighter bounds, better constructions, and completely different approaches.

We welcome:

(1) Alternative formalizations that capture different aspects of the problem;

(2) Challenges to our assumptions and theoretical bounds;

(3) Completely different approaches we have not considered;

(4) Identification of gaps or weaknesses in our framework;

(5) Extensions and refinements from any research community;

The measure of success for foundational work is not being right about everything, but catalyzing important research. If our formulation is eventually superseded by better frameworks, or if solutions emerge through approaches we never imagined, we would consider that success. Every perspective enriches our collective understanding. We see critique not as opposition but as collaboration toward solving a shared challenge.

We sincerely look forward to more discussion around the OML primitive, and would like to, once again, extend our deepest gratitude to all the reviewers for engaging in the fruitful discussion with us.

---

### Meta-Review · Area_Chair_Bhs6 · 2025-12-11

**Summary:**

The paper formulates the problem that model owners still want to execute control on their models, even when these models are distributed. Therefore, as a solution, the paper formulates OML which enables Open-access, Monetizability, and Loyalty.

The reviewers formulated strong concerns regarding the contribution:
1.  The post-hoc mechanism cannot satisfy the pre-hoc Loyal property. The two potential solutions (i.e., providing a a concrete pre-hoc verifier pathway, or removing the formulation) have not been implemented.
2. The lack of a final sample bound.
3. The lack of a formal game-theoretic model for the Nash equilibrium.
4. Lack of novelty, especially for the security community.
5. The absence of a true open-access policy.
6. The vague description of the main contribution in the main text.

**Reviewer Concerns:**

While the rebuttal agreed with all the reviewers' concerns, it did not address them but just argued about them, and promised future updates to the paper. The reviewers explicitly formulate that their concerns have not been addressed through this rebuttal (but partially been reinforced).

**Reviewer Scores:**

It sounded like they would have maintained their current scores.

---

### Decision · Program_Chairs · 2026-01-26

Reject